# Nucleosome-directed replication origin licensing independent of a consensus DNA sequence

Sai Li[1], Michael R. Wasserman[1,7], Olga Yurieva[2,3], Lu Bai [4,5,6], Michael E. O'Donnell[2,3] ✉ & Shixin Liu [1] ✉

The numerous enzymes and cofactors involved in eukaryotic DNA replication are conserved from yeast to human, and the budding yeast *Saccharomyces cerevisiae* (S.c.) has been a useful model organism for these studies. However, there is a gap in our knowledge of why replication origins in higher eukaryotes do not use a consensus DNA sequence as found in S.c. Using in vitro reconstitution and single-molecule visualization, we show here that S.c. origin recognition complex (ORC) stably binds nucleosomes and that ORC-nucleosome complexes have the intrinsic ability to load the replicative helicase MCM double hexamers onto adjacent nucleosome-free DNA regardless of sequence. Furthermore, we find that *Xenopus laevis* nucleosomes can substitute for yeast ones in engaging with ORC. Combined with re-analyses of genome-wide ORC binding data, our results lead us to propose that the yeast origin recognition machinery contains the cryptic capacity to bind nucleosomes near a nucleosome-free region and license origins, and that this nucleosome-directed origin licensing paradigm generalizes to all eukaryotes.

Complete and accurate genome duplication prior to cell division is a critical process for essentially all living organisms. The basic mechanism for the initiation of DNA replication is shared across domains of life[1,2]. An "initiator" first binds to genomic sites referred to as origins of replication, then recruits the replicative helicase responsible for unwinding parental duplex DNA, creating two complementary templates for DNA polymerases to form two identical copies of daughter chromosomes. In eukaryotes, multiple origins across the genome are licensed for firing once, and only once, per cell cycle and their firing follows a temporally controlled program[3–5]. The eukaryotic initiator is known as the origin recognition complex (ORC), which consists of six conserved subunits, Orc1-6[6,7]. ORC works in concert with Cdc6 and Cdt1 to coordinate the loading of two Mcm2-7 helicase complexes (MCM) onto the origin DNA[8,9], forming the MCM double hexamer

(DH), or "pre-replication complex" (pre-RC). Recent reports demonstrate that the two MCM hexamers are loaded onto DNA by two ORCs or by one ORC that flips between two elements within the origin[10–13]. Only a fraction of MCMs loaded on chromatin become activated during the subsequent S phase to produce bidirectional replication forks. The excess of chromatin-associated MCM over those used for replication is referred to as the "MCM paradox"[14].

In the budding yeast *Saccharomyces cerevisiae* (S.c.), origins are located at a set of "replicator" positions known as autonomously replicating sequences (ARS), each containing a 17-bp AT-rich ARS consensus sequence (ACS) element and other less conserved "B elements"[1]. The ARS sequence is AT rich which is not conducive to nucleosome assembly, forming a "nucleosome-free region" (NFR). An NFR at an ARS is important for origin function[15], and when

[1]Laboratory of Nanoscale Biophysics and Biochemistry, The Rockefeller University, New York, NY, USA. [2]Laboratory of DNA Replication, The Rockefeller University, New York, NY, USA. [3]Howard Hughes Medical Institute, The Rockefeller University, New York, NY, USA. [4]Center for Eukaryotic Gene Regulation, The Pennsylvania State University, University Park, PA, USA. [5]Department of Biochemistry and Molecular Biology, The Pennsylvania State University, University Park, PA, USA. [6]Department of Physics, The Pennsylvania State University, University Park, PA, USA. [7]Present address: Syros Pharmaceuticals, Cambridge, MA, USA. ✉e-mail: odonnel@rockefeller.edu; shixinliu@rockefeller.edu

nucleosomes are permitted to encroach upon the ARS they are demonstrated to be inhibitory[16–18]. While S.c. has been an instrumental model organism to study eukaryotic replication initiation, the use of a consensus origin DNA sequence is only found in some budding yeasts[19], and therefore a consensus origin sequence is the exception rather than the norm in eukaryotes[19,20]. For example, defined origin elements are not present in the fission yeast *Schizosaccharomyces pombe*[21]. The specificity of S.c. ORC for ARS is due to a unique insertion helix (IH) in the Orc4 subunit, which specifically interacts with the ACS sequence[22–25]. S.c. strains with Orc4 IH mutations alter genome-wide firing patterns[23,24] and may enable initiation in NFRs that are different from ARS, such as at transcriptional promotors. It is also worth noting that ARS and its internal ACS and B elements are not strictly required in vitro for S.c. ORC binding or MCM loading[9,26]. These findings, combined with the lack of ACS-like elements in higher eukaryotes, indicate that there exist other chromosomal features that enable the licensing and firing of eukaryotic replication origins[19,27].

One distinct challenge for ORC is the need to navigate through chromosomes prevalently packaged into nucleosomes[28], which influence multiple aspects of eukaryotic replication[29–31]. The nucleosome is often viewed as a barrier to ORC binding and origin activity that must be overcome by certain DNA sequences or chromatin-remodeling enzymes[16–18]. ORC is known to directly interact with nucleosomes by multiple connections[32,33]. Given the continuing uncertainties of the various contacts of ORC subunits to nucleosomes, the interplay between ORC and nucleosomes and its role in origin selection and function remain an area of active research.

In the current study we unexpectedly discovered that budding yeast contain a cryptic nucleosome-dependent mechanism for origin licensing that is independent of an ARS consensus, implying its

capability to license origins independent of DNA sequence, and suggesting this process may provide a general mechanism for origin selection that applies to all eukaryotes. On hindsight, this is supported by an extensive ChIP-seq study that determined about one-third of S.c. origins lack a recognizable consensus ACS sequence[15]. Based on the facts that (1) ORC binds nucleosomes[32,33], and (2) pre-RCs are predominantly licensed at NFRs[15,16], we hypothesize that nucleosomes recruit ORC and that this can lead to assembly of the MCM DH onto DNA provided, at a minimum, that there is an adjacent NFR that can accommodate ORC binding and MCM loading (Fig. 1). We presume that there may be additional requirements for origin activation in vivo at these "non-ARS" sites, such as histone modifications or nucleosome remodelers.

Why would S.c. have developed sequence-specific origins and also utilize a general nucleosome-directed but non-sequence-specific method for origin selection? We presume S.c. evolved ARS sequences, located at intergenic regions, for preferred ORC binding due to its small genome size and therefore urgent need to avoid transcription-replication conflicts, as proposed earlier[24]. To test whether origin licensing in S.c. can occur at nonspecific DNA sequence that is adjacent to a nucleosome, we develop herein a single-molecule platform to visualize the dynamic behavior of ORC and MCM on nucleosomal DNA. We find that S.c. ORC stably associates with nucleosomes and loads MCM DHs at adjacent NFRs. This finding of ARS sequence-independent yet nucleosome-directed ORC binding and subsequent MCM DH loading reveals that the S.c. system has an inherent sequence-independent origin licensing activity that is akin to that in higher eukaryotes, thereby providing insight and a unifying mechanism for eukaryotic origin selection.

## Results

### A single-molecule platform to directly visualize eukaryotic origin licensing

To perform single-molecule studies on pre-RC formation, we used the bacteriophage λ genomic DNA containing an engineered S.c. ARS1 sequence[34] placed 14 kb from one end (Fig. 2a and Supplementary Fig. 1). The DNA construct (termed λ$_{ARS1}$) was biotinylated at both ends. We expressed and purified the S.c. Orc1-6 and Mcm2-7 complexes (referred to as ORC and MCM hereafter) as well as Cdc6 and Cdt1 (Supplementary Fig. 2). To generate fluorescently labeled ORC for direct visualization, we site-specifically attached a Cy3 fluorophore to the N terminus of the Orc1 subunit via a 12-residue S6 peptide tag. A single λ$_{ARS1}$ DNA molecule was tethered between a pair of streptavidin-coated beads in the microfluidic chamber of a dual-trap optical tweezers instrument combined with multicolor confocal fluorescence microscopy[35,36] (Fig. 2b). Upon moving the tethered DNA into a channel containing Cy3-ORC (±Cdc6) and ATP, we observed ORC binding to DNA in real time. In the presence of Cdc6, ORC displayed short-lived and diffusive binding to non-ARS1 DNA, while remaining stably bound at the ARS1 site (Fig. 2c). Cdc6 enhances the overall binding of ORC to DNA and is required for its ARS specificity (Fig. 2d, e). These results are consistent with previous single-molecule studies[10,37], thus confirming the normal function of proteins used in this study. The distinctive behavior of ORC at the engineered ARS1 site versus all other DNA sites indicates that the λ$_{ARS1}$ template does not contain a second strong ACS motif, which is corroborated by sequence analysis of the λ genomic DNA (Supplementary Fig. 3).

### ORC mediates MCM DH loading onto both ARS and non-ARS DNA

Next we generated a S.c. strain expressing Mcm2-7 with S6-tagged Mcm3, enabling us to site-specifically label MCM complexes with a fluorophore. We used LD650-labeled MCM, along with unlabeled ORC, Cdc6 and Cdt1, to examine pre-RC formation on λ$_{ARS1}$ DNA (Fig. 3a). Surprisingly, we observed that the majority (86%) of MCM that stably

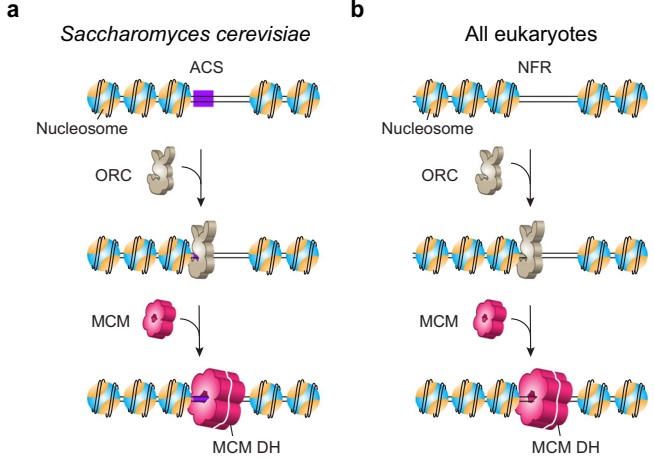

**Fig. 1 | Model for origin licensing in *Saccharomyces cerevisiae* vs. in most other eukaryotes. a** In a chromatinized genome, ORC searches for nucleosomes and stably associates with them regardless of whether a nearby ARS consensus sequence (ACS) exists. ORC then loads MCM double hexamers in conjunction with Cdc6 and Cdt1 at nucleosomal sites in G1 phase. Due to its small genome, *S. cerevisiae* may have evolved ARS-dependent origins in order to limit ORC binding to specific sequences and thus avoid replication-transcription conflicts[24], which does not generalize to higher eukaryotes. **b** Origins in most eukaryotes lack a consensus ACS motif, but still utilize ORC which is known to bind nucleosomes. We demonstrate in this study that S.c. ORC harbors a cryptic ability to bind nucleosomes and direct MCM DH formation within nucleosome-free regions (NFRs) that lack ARS consensus sequences. We propose that this mechanism is the normal process for most eukaryotes, whose origin sites are chiefly defined by the nucleosomal architecture. In this model, ACS confers origin specificity in S.c. by facilitating ORC binding to nucleosomes proximal to a nucleosome-free ARS sequence. We note that MCM DHs are recruited to origins via a "single ORC flipping" or "two ORC" mechanism[10–13], and that our model is compatible with either scenario.

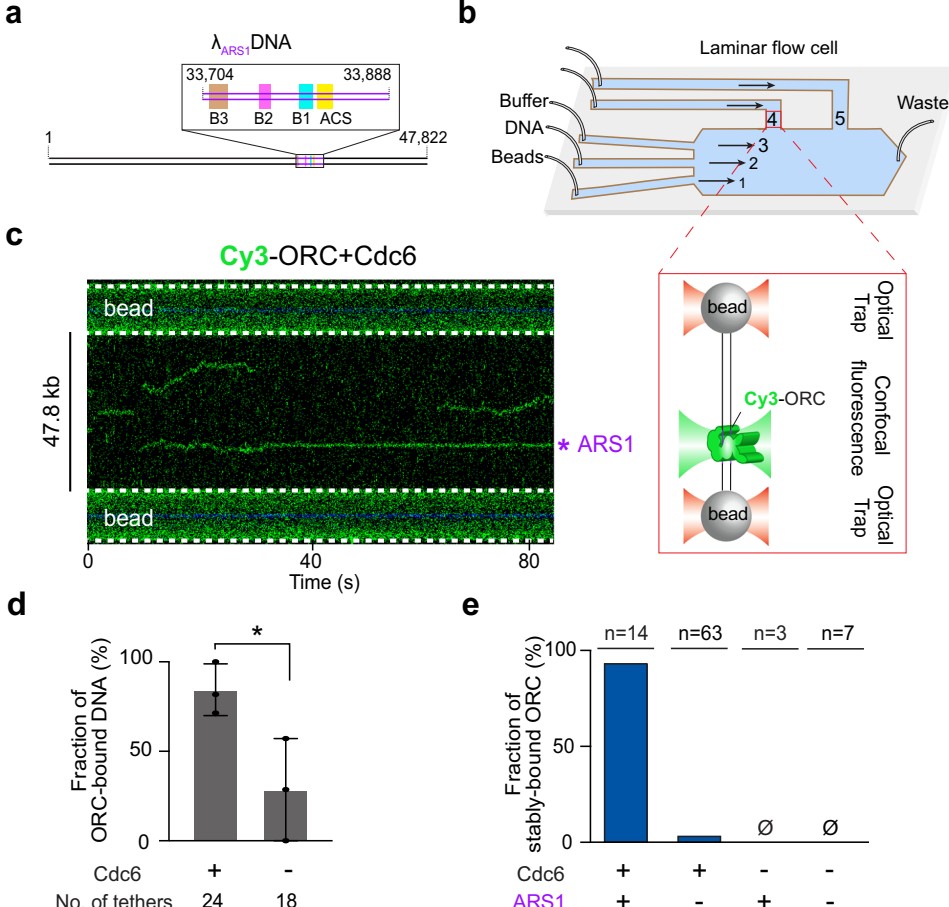

**Fig. 2 | A single-molecule platform to study eukaryotic replication initiation.**
**a** Cartoon of the $\lambda_{ARS1}$ DNA template. The inserted ARS1 element is illustrated in the inset box. **b** Schematic of the single-molecule experimental setup. Channels 1–3 are separated by laminar flow. Beads are optically trapped in channel 1, moved to channel 2 to tether DNA, then moved to channel 3 to characterize the force-extension curve of the tether. Once a correct tether is confirmed, the beads-DNA assembly is moved to channel 4 or 5 containing the proteins. The zoom-in box illustrates the final assembly in the imaging channel (not drawn to scale). **c** A representative kymograph showing the behavior of Cy3-labeled ORC on $\lambda_{ARS1}$ DNA in the presence of Cdc6. The engineered ARS1 position is indicated. **d** Fraction of

$\lambda_{ARS1}$ DNA tethers that were observed to have at least one ORC bound in the presence or absence of Cdc6. The protein concentrations used in this experiment are: 2 nM for ORC, and 5 nM for Cdc6. The number of tethers analyzed for each condition is indicated. Data are presented as mean values ± SD from three independent experiments. Significance was obtained using an unpaired two-tailed $t$-test ($*p < 0.05$). **e** Fraction of ORC molecules that stably reside at the ARS1 site vs. non-ARS1 sites in the presence or absence of Cdc6. $n$ indicates the number of ORC molecules analyzed for each condition. Source data are provided as a Source Data file.

bound to DNA were at positions distant from ARS1 (Fig. 3b, c). Considering that non-ARS DNA sites vastly outnumber the sole ARS1 site in our template, ORC/MCM still appears to have a higher affinity to ARSs than to other DNA sequences. We then conducted photobleaching analysis to examine the stoichiometry of MCM bound to DNA, and observed that a significant fraction (36%) of MCMs underwent two-step photobleaching (Fig. 3d, e). Given the estimated labeling efficiency for MCM (~60%), this result suggests that the majority of MCMs on DNA observed in our experiments were DH. The presence of ORC is strictly required for MCM binding to DNA (Fig. 3f). Notably, we have shown that ORC displays diffusive behavior at non-ARS sites (Fig. 2c, e). Therefore, it appears that MCM stabilizes ORC binding to non-ARS DNA sequences.

To obtain further evidence for MCM DH formation, we used a mixture of Cy3-labeled and LD650-labeled MCMs. After ORC-mediated MCM binding, we moved the DNA tether into a separate channel containing a high-salt buffer (0.5 M NaCl), upon which we observed high mobility of dual-colored MCM (red and green) (Fig. 3g, h). We found a significant portion of the MCM complexes remained associated with DNA upon high-salt wash—either stably residing at the initial position or sliding along the DNA (Fig. 3i), consistent with the

behavior of properly loaded MCM DHs[8–10,38]. The rest of the MCMs that dissociated from DNA at high salt likely represent those that did not topologically encircle the DNA duplex. The fraction of loaded MCMs at the ARS1 site (62%) is much higher than that at non-ARS sites (25%). Therefore, our results confirm that ARS1 represents a preferred position for ORC-mediated MCM loading, but also unambiguously show that this process can still occur on non-ARS sequences. In other words, our data suggest that ARS enhances the likelihood, but is not strictly required, for ORC to license an origin.

## ORC preferentially binds nucleosomes over nucleosome-free DNA

Next, we set out to examine the behavior of ORC on chromatinized DNA. We reconstituted both S.c. and *Xenopus laevis* (X.l.) histone octamers for comparative studies (Supplementary Figs. 2 and 4). In both cases, a unique cysteine residue is placed on the histone H2A, enabling site-specific labeling of the nucleosome. We used the histone chaperone Nap1 for in situ nucleosome assembly[30,39]. By titrating the protein concentrations, we found a condition that yields sparsely populated nucleosomes (usually between 1 and 7) within the tethered $\lambda_{ARS1}$ DNA, such that most if not all of the nucleosomes are flanked by

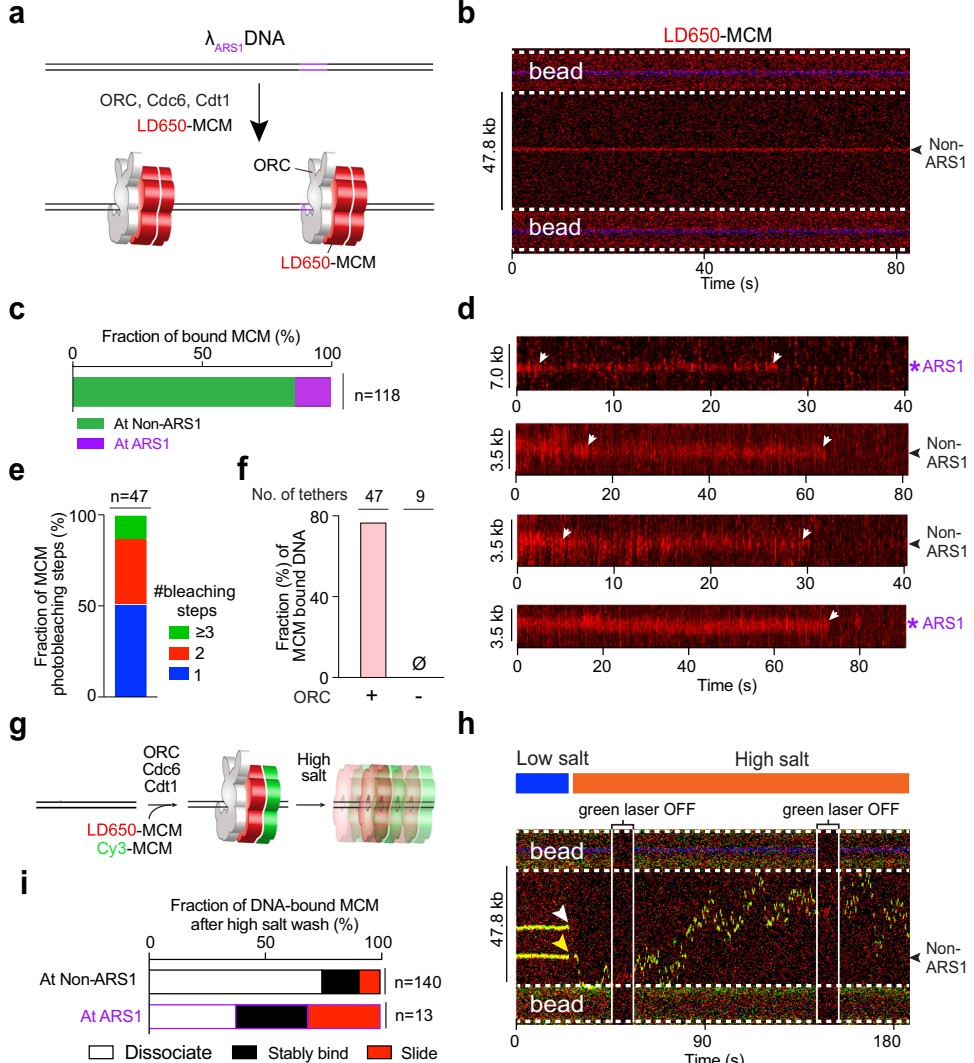

**Fig. 3 | ORC-dependent MCM loading occurs frequently at non-ARS DNA sites.**
**a** Cartoon of the single-molecule pre-RC assembly experiment using λ_ARS1 DNA, unlabeled ORC, Cdc6, Cdt1, and LD650-labeled MCM (red). **b** An example kymograph showing that the MCM fluorescence signal appeared at a non-ARS1 position on DNA. **c** Fraction of stably bound MCM complexes that were observed at ARS1 vs. non-ARS1 positions. *n* indicates the number of MCM complexes analyzed.
**d** Example kymographs showing the photobleaching steps (white arrows) of MCM fluorescence at ARS1 and non-ARS1 positions on the DNA tether. **e** Distribution of the number of photobleaching steps observed in each MCM fluorescence trajectory. *n* indicates the number of trajectories analyzed. **f** Fraction of DNA tethers that were observed to harbor at least one fluorescent MCM complex in the presence or absence of ORC. The protein concentrations used in this experiment are: 10 nM for

MCM, 2 nM for ORC, and 5 nM for Cdc6. The number of tethers analyzed for each condition is indicated. **g** Cartoon of the high-salt wash experiment to demonstrate MCM loading on DNA using a mixture of LD650-MCM and Cy3-MCM, unlabeled ORC, Cdc6 and Cdt1. **h** A representative kymograph showing large-scale mobility of an MCM DH (indicated by the dual-color complex which appeared as yellow) loaded at a non-ARS1 position traversing the entire length of the tethered DNA upon high-salt wash (yellow arrowhead). The other MCM complex dissociated at high salt (white arrowhead). **i** Fraction of MCM complexes on nucleosome-free DNA that underwent sliding on DNA without dissociation (red), remained stably bound to the DNA position (black), or dissociated into solution (white) upon high-salt wash. *n* indicates the number of MCM complexes analyzed. Source data are provided as a Source Data file.

substantial NFRs (Fig. 4a and Supplementary Fig. 5). When incubating the DNA tether harboring Cy3-labeled S.c. nucleosomes with LD650-ORC and Cdc6, we made the striking observation that ORC predominantly colocalized with the nucleosomes, rather than residing within the long stretches of bare DNA between nucleosomes (Fig. 4b). Importantly, stable association of ORC with the nucleosome is independent of whether the nucleosome is located at the ARS1 position or at non-ARS sites (Fig. 4c). In fact, most nucleosomes were located at non-ARS positions and yet the vast majority (>90%) had a stably bound ORC using only a low concentration (2 nM) of ORC (Fig. 4d). Sometimes we observed ORC at the nucleosome-free ARS1 DNA position where it remained stably bound in the presence of Cdc6 (e.g., second kymograph in Supplementary Fig. 6a), again supporting the expected and normal ORC function. In contrast, when ORC bound to non-ARS

bare DNA within a nucleosomal DNA tether, it displayed diffusive behavior and the diffusion was confined between adjacent nucleosomes that were occupied by ORC (Fig. 4b and Supplementary Fig. 6).

Comparing Fig. 2e (ORC behavior on non-nucleosomal DNA) and Fig. 4c (ORC behavior on nucleosomal DNA), it becomes apparent that the presence of nucleosomes on DNA abrogates the requirement of ARS sequences for stable ORC binding. In other words, ORC has the ability to stably engage with a nucleosome regardless of its adjacent DNA sequence. Interestingly, stable association of ORC with nucleosomes appears to be a conserved phenomenon as we obtained similar results using either S.c. or X.l. histone octamers (Fig. 4c, d). Nucleosome targeting by ORC can conceivably be achieved by either a three-dimensional (3D) search (direct binding from solution) or a one-dimensional (1D) search (sliding along the DNA from a non-

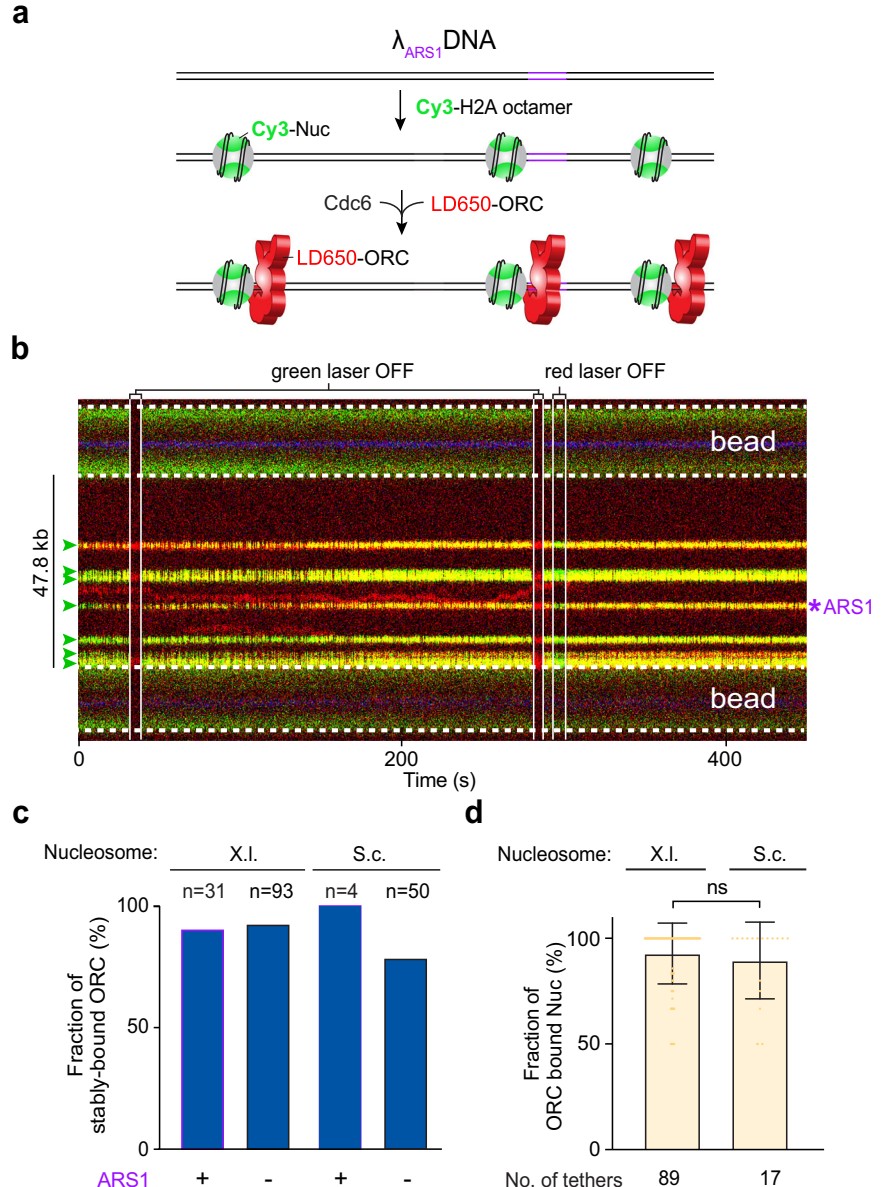

**Fig. 4 | ORC predominantly binds to nucleosomes over bare DNA. a** Cartoon of the $\lambda_{ARS1}$ DNA sparsely loaded with Cy3-labeled S.c. nucleosomes (green) and incubated with LD650-labeled ORC (red) and Cdc6. **b** A representative kymograph showing a $\lambda_{ARS1}$ DNA tether loaded with multiple nucleosomes (positions indicated by green arrowheads), all of which were located at non-ARS1 sites except one. Each nucleosome was observed to be stably bound by ORC (red). The presence of ORC on the nucleosomes is confirmed by turning off the green laser, which showed only the red fluorescence from ORC; alternatively, turning off the red laser showed only the green fluorescence from the nucleosomes. **c** Fraction of ORC stably bound to nucleosomes (X.l. or S.c.) located at either the ARS1 site or non-ARS1 sites. *n* indicates the number of ORC molecules analyzed for each condition. **d** Fraction of nucleosomes (X.l. or S.c.) within a given DNA tether that were observed to be ORC-bound in the presence of 2 nM ORC and 5 nM Cdc6. The number of tethers analyzed for each condition is indicated. Data are presented as mean values ± SD. Significance was obtained using an unpaired two-tailed *t*-test (ns, *p* = 0.41). Source data are provided as a Source Data file.

nucleosomal site). We observed both modes in our data, with 3D search being the dominant mode, especially for S.c. nucleosomes (Supplementary Fig. 6).

## ORC mediates MCM loading to nucleosomes

Next we asked whether ORC binding to nucleosomes can lead to MCM helicase loading and pre-RC formation. We used fluorescently labeled MCM complexes (with unlabeled ORC, Cdc6, and Cdt1) to examine their recruitment to nucleosomal DNA and its dependence on ORC. We found that MCM frequently colocalized with nucleosomes that were sparsely distributed across the DNA tether (Fig. 5a, b). This observation was made for both S.c. and X.l. nucleosomes. MCM binding to the nucleosome requires the presence of ORC, as omitting ORC completely eliminated MCM-nucleosome colocalization (Fig. 5C). Considering that the free DNA sites vastly outnumber the nucleosome sites in our assay (a few nucleosomes within 48-kbp DNA), the observed frequency of MCM-nucleosome colocalization indicates that nucleosomes are preferred sites for ORC-mediated MCM binding (Fig. 5d). Again, ARS sequences are not required for MCM-nucleosome colocalization, as these events were mostly observed at non-ARS sites (Fig. 5e). To examine whether MCM can form the DH at the nucleosome, we performed three-color fluorescence experiments using A488-labeled histone octamers and a mixture of Cy3- and LD650-labeled MCMs (Fig. 5f). Indeed, we detected colocalization of dual-color MCMs (green and red) with nucleosomes (blue), suggesting MCM DH formation at nucleosome sites.

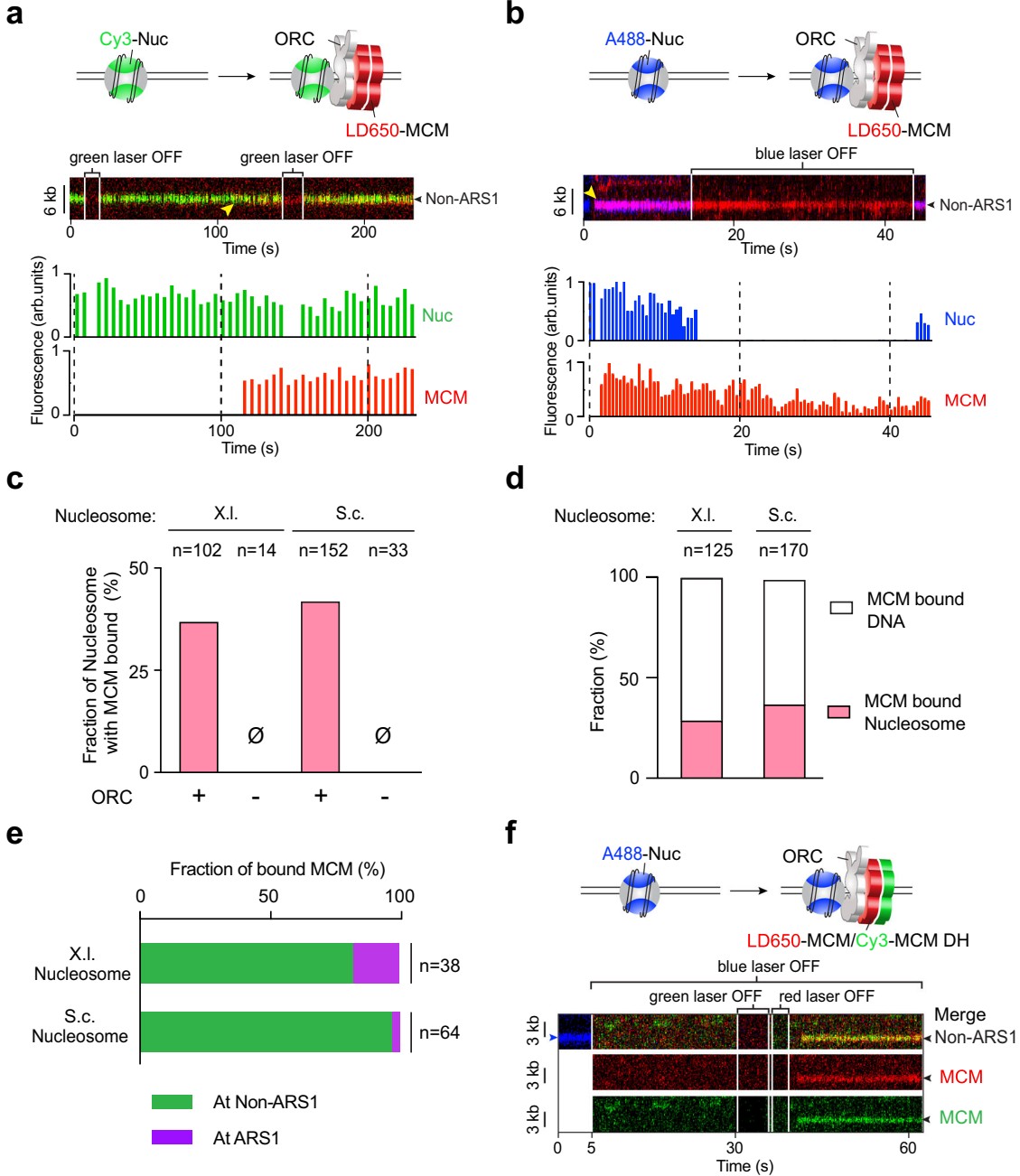

**Fig. 5 | MCMs are recruited by ORC to nucleosomes independently of ARS DNA.**
**a** Cartoon (top), an example kymograph (middle) and the corresponding fluorescence intensities (bottom) of the pre-RC assembly experiment using Cy3-labeled X.l. nucleosomes (green), LD650-labeled MCM (red), unlabeled ORC, Cdc6 and Cdt1. Yellow arrowhead in the kymograph indicates the time when the MCM fluorescence signal appeared at the nucleosomal site. **b** Cartoon (top), an example kymograph (middle) and the corresponding fluorescence intensities (bottom) of the pre-RC assembly experiment using A488-labeled S.c. nucleosomes (blue), LD650-labeled MCM (red), unlabeled ORC, Cdc6 and Cdt1. In both examples in **a**, **b** the nucleosomes were at non-ARS1 positions on the DNA. **c** Fraction of nucleosomes (X.l. or S.c.) that were observed to have colocalized MCM signals in the presence or absence of ORC. *n* indicates the number of nucleosomes analyzed

for each condition. **d** Fraction of MCM complexes on a nucleosome-loaded (X.l. or S.c.) tether that colocalized with a nucleosome vs. with nucleosome-free DNA. *n* indicates the number of MCM complexes analyzed. **e** Fraction of MCM-nucleosome (X.l. or S.c.) colocalization events observed at ARS1 vs. non-ARS1 positions. *n* indicates the number of events analyzed. **f** Cartoon (top) and an example kymograph (bottom) of the three-color experiment using A488-labeled S.c. nucleosomes (blue), both LD650-labeled MCM (red) and Cy3-labeled MCM (green), unlabeled ORC, Cdc6 and Cdt1. The colocalization of a dual-color MCM with a nucleosome indicates MCM DH recruitment to the nucleosomal site. Individual lasers were occasionally turned off to confirm the fluorescence signals from the other channels. Source data are provided as a Source Data file.

To test whether MCM DHs are truly loaded (i.e., topologically encircling DNA), we moved the tethered nucleosomal DNA with bound ORC and MCM into a high-salt buffer channel containing 0.5 M NaCl. As explained earlier, recruited MCMs that do not encircle DNA are expected to dissociate at high salt, whereas loaded MCM DHs are

expected to stay on DNA and can slide on it if ORC is dislodged by high salt[8,9]. Notably, we found that, unlike ORC removal from free DNA (ARS1 or non-ARS1) by high salt in the absence of nucleosomes, MCM-ORC complexes appeared more resistant to salt when bound to nucleosomes. Thus we used labeled ORC to examine the ability to

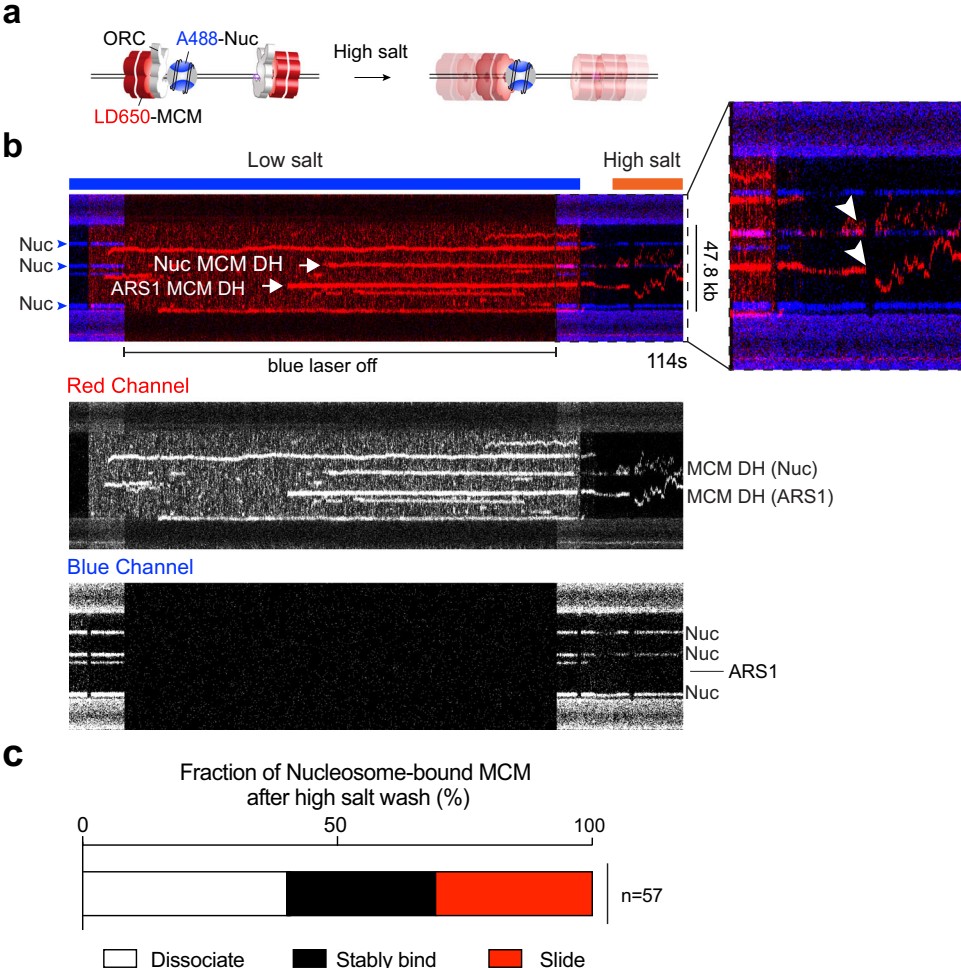

**Fig. 6 | ORC-mediated MCM loading occurs at nucleosomal sites. a** Cartoon illustrating the experimental assay that evaluates MCM loading at nucleosomal sites via high-salt wash. **b** A representative kymograph showing that MCM complexes (red) formed on a DNA tether in the presence of unlabeled ORC, Cdc6 and Cdt1. Upon moving to a high-salt buffer (0.5 M NaCl), a fraction of the MCMs displayed diffusive behavior without dissociation, demonstrating their successful loading onto DNA. MCM diffusion could occur from both bare DNA and nucleosome sites (white arrowheads in the zoomed-in view). Blue arrowheads indicate nucleosome positions, all of which were at non-ARS1 sites in this example. MCM and nucleo-some fluorescence signals are also separately shown in gray scale at the bottom. **c** Fraction of nucleosome-colocalized MCM complexes that underwent diffusion without dissociation (red), remained stably bound to the nucleosome (black), or dissociated into solution (white) upon high-salt wash. *n* indicates the number of MCM complexes analyzed. Source data are provided as a Source Data file.

dissociate ORC from nucleosomes at 0.5 M NaCl. With both types of nucleosomes (S.c. and X.l.), we found that a significant fraction of ORC was retained in 0.5 M NaCl (Supplementary Fig. 7). While it appears that S.c. ORC is more frequently retained at X.l. nucleosomes than at S.c. nucleosomes, this could be due to subtle differences in the experimental conditions rather than a meaningful difference in their binding affinity.

We then analyzed the behavior of MCM recruited to ORC-bound S.c. nucleosomes upon high-salt wash. The example kymograph in Fig. 6a shows multiple fluorescently labeled MCMs (red) on tethered DNA. Upon high-salt wash, the MCM colocalized with a non-ARS positioned nucleosome (blue) underwent sliding on DNA, as did another MCM located at the ARS1 DNA site without a nucleosome. Another kymograph in Supplementary Fig. 8a–c shows dual-color MCM DH sliding at high salt. Overall we observed ~30% of MCMs formed at ORC-nucleosome sites to diffuse on DNA and another ~30% to remain associated with the nucleosome (Fig. 6b and Supplementary Fig. 8d). The immobile MCM fraction can be attributed to the strong engagement of ORC with the nucleosome (Supplementary Fig. 7), which in turn holds MCM next to the nucleosome. The remainder of the MCMs dissociated into solution upon high-salt wash, which presumably represent MCMs that were not fully loaded onto DNA. We note that the MCM behavior at nucleosomes reported in Fig. 6c shows a very similar distribution to the one for ARS1 DNA (Fig. 3i). While a rigorous proof needs further investigation, this similarity in high-salt resistance indicates that MCM DHs at ARS1 DNA and at nucleosomes are loaded through the same process and are functionally equivalent.

**Genome-wide analysis of ORC binding and ACS motifs**
The single-molecule results presented in this study suggest that ORC binding and MCM loading may occur over genomic regions that lack an ACS element. In support of this observation, it has been previously shown that only two-thirds of well-documented ARS sites in the S.c. genome contain a recognizable ACS sequence[15]; while the remaining third were presumed to contain "novel ACS sequences". We propose, based on the work presented here, that the novel ACS sequences may have been random DNA sequences adjacent to a nucleosome at an NFR.

To test this hypothesis, we re-analyzed published S.c. Orc1 and Mcm2-7 ChIP-seq data[15,40]. Consistent with previous analysis[15], we identified 295 Orc1 ChIP peaks genome-wide, and most of these sites also bind to Mcm2-7 (Fig. 7a). Most of the Orc1 peaks (225 out of 295)

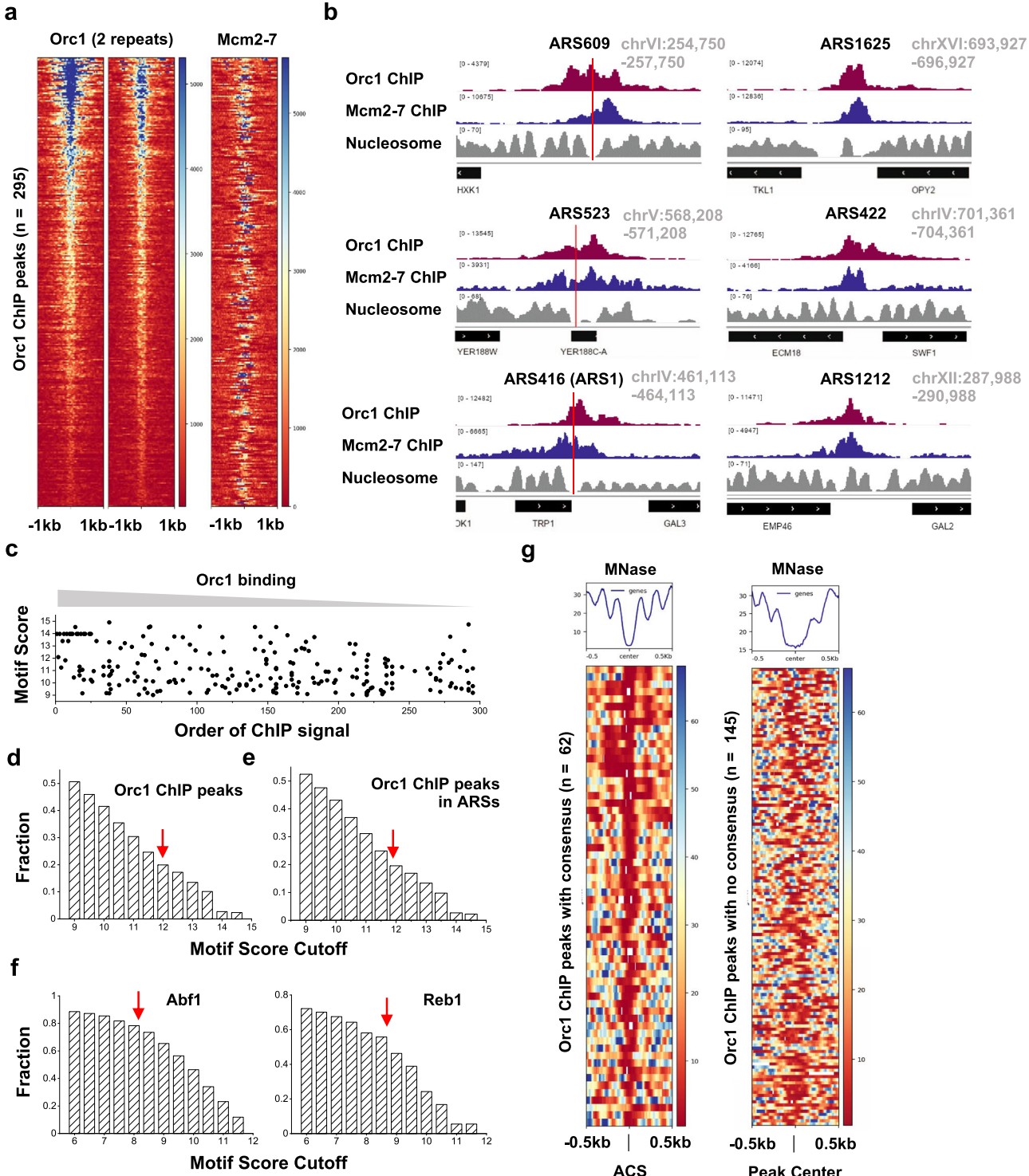

**Fig. 7 | Genome-wide analysis of ORC/MCM localization and ACS motif scores.**
**a** Heatmap of Orc1 ChIP peaks (*n* = 295) and Mcm2-7 ChIP signal at the corresponding sites. **b** Examples of Orc1 and Mcm2-7 ChIP data and nucleosome occupancy at six ARS. Red vertical lines represent the location of ACS consensus in these regions. The three ARS on the right contain no consensus with score above 9. **c** ACS motif score vs. Orc1 ChIP peak strength. The motifs were identified in a 1-kb region near Orc1 ChIP peaks (peak center ±500 bp). All elements with a PWM score >9 are shown here. For the *x* axis, "1" represents the largest Orc1 ChIP peak, and "295" the weakest. **d** Fraction of sequences underlying all Orc1 peaks that contain motifs above a certain threshold (varying from 9 to 15). The red arrow represents the recommended cutoff of 11.9. **e** Same as in **d** except using a subset of Orc1 peaks that overlap with previously annotated ARS. **f** Abf1 and Reb1 ChIP peaks analyzed in the same way as in **d**. The red arrows represent recommended cutoffs for these two factors. **g** Heatmap of nucleosome occupancy near Orc1 ChIP peaks. The left panel includes 62 peaks containing ACS consensus (PWM > 11.9), with each row aligned at the consensus site. The right panel includes 145 peaks with no consensus above 9, and it was aligned at the center of the ChIP peaks.

overlap with annotated origins of replication[41]. Examples of ORC/MCM peaks at known origin sites are shown in Fig. 7b. We extracted DNA sequences ±500 bp flanking the center of the ChIP peaks, scanned each 17-bp window with the ACS position weight matrix (PWM)[42], and recorded all the windows with a PWM score of >9 (previous studies recommended a cutoff score of 11.9). We found that, even with this less stringent criterion, 145 out of the 295 Orc1 peaks contain no qualified window, 105 with one qualified window, and the remaining 45 with multiple qualified windows. The PWM scores are not correlated with the strength of the ChIP peaks (Fig. 7c). For example, among the six examples shown in Fig. 7b, ARS609, ARS523, and ARS416 contain an ACS with high-to-medium scores of 13.2, 14.0, and 10.1, respectively, while the rest contain no window with a score above 9. Yet all of these sites show comparable Orc1 and Mcm2-7 ChIP signals. This result suggests that ORC/MCM binding to established S.c. origins does not require a highly consensus ACS.

To rule out the possibility that the above conclusion depends on the choice of the cutoff score, we calculated the fraction of origin sequences that contain at least one qualified window using different cutoff values (Fig. 7d). As expected, fewer qualified sequences were identified with a higher cutoff ACS PWM score. Using the conventional cutoff of 11.9, the ACS consensus is only identified in ~20% of Orc1 ChIP peaks. Among the 225 Orc1 peaks that overlap with annotated origins, we observed a similar fraction (~20%) (Fig. 7e). This value only mildly increases to ~25% when we analyzed the subset of Orc1 ChIP peaks that also have high Mcm2-7 ChIP signals (Supplementary Fig. 9a, b). To test the above conclusion further, we also examined the recent Orc1 ChIP-exo data[43]. Although only 81 Orc1 peaks were identified in this dataset, the probability of finding a consensus ACS motif in these peaks is essentially the same (~20% with a cutoff score of 11.9) (Supplementary Fig. 9c). As a quality control of our method, we applied the same analysis to sequence-specific transcription factors Abf1 and Reb1, and found that their cognate DNA motifs occur at a much higher frequency near their respective ChIP peaks (~75% and ~55% at recommended cutoff of 8.2 and 8.7, respectively) compared to Orc1 (Fig. 7f). Overall, these results suggest that a significant fraction of established replication origins in the yeast genome do not contain a highly consensus ACS.

### Genome-wide analysis of ORC binding and NFRs

We next investigated the location of Orc1 ChIP peaks relative to the neighboring nucleosomes. Nucleosome occupancy measured by MNase assay is also shown for the examples in Fig. 7b [44]. For the first three origins with a well-defined ACS, the consensus sequence all locates inside NFRs (indicated by the red lines), in accord with the fact that ACS motifs are AT-biased sequences that generally exclude nucleosomes[45]. Indeed, when we collected all 62 Orc1 peaks with a consensus ACS (PWM score >11.9), aligned at their ACS locations, and plotted the nearby nucleosome occupancy, we observed strong nucleosome depletion over most of the sites (Fig. 7g). This is consistent with the previous finding that a consensus ACS engineered into a well-positioned nucleosome leads to nucleosome displacement[46]. On the other hand, Orc1 ChIP peaks over sequences without a consensus ACS motif (PWM score <9) also contain NFRs (Fig. 7b). In these cases, ORC potentially bind to nucleosome-NFR junctions, although higher resolution data are needed to resolve the exact ORC binding location relative to nucleosomes.

There are four to five thousand NFRs in the yeast genome[47], but fewer than 300 Orc1 binding sites, indicating that most NFRs do not serve as origins. By analyzing Orc1 ChIP peaks and ACS motifs in genome-wide NFRs, we found that the ACS motif plays a strong role in directing ORC binding only when it is a near-perfect match to the consensus. For example, nearly 90% of ACS motifs with a score >14 are occupied by Orc1 (Supplementary Fig. 10a, b). For weak motifs, the correlation between ORC binding and the motif score is very low (Supplementary Fig. 10c).

## Discussion

In this work, we present findings that support a model in which S.c. ORC either binds to consensus ACS elements in the genome—which may lead to NFR formation—or binds to nucleosomes near an NFR (Fig. 1), indicating an unexpected level of flexibility in yeast origin architecture[48].

### Nucleosomes play a major role in directing ORC function in origin licensing

The prevailing model for eukaryotic origin selection takes a DNA-centric view, which argues that ARS—especially the ACS element within ARS—largely dictates where ORC binds in the genome[1]. Nonetheless, in vitro studies demonstrate that these elements are not strictly required for pre-RC formation or replication initiation even for the yeast system where ARSs were initially identified[6,9,26]. For example, the in vitro MCM loading efficiency was similar between wildtype ARS1 and ACS-deleted ARS1, and the specificity of MCM DH loading at ARS1 required addition of competitor DNA[9]. Moreover, the yeast genome contains far more ACS motifs than functional origins[49]. Therefore, the determinants of eukaryotic origin selection must include other chromosomal features, the identity of which are still under study. By imaging ORC's behavior on nucleosomal DNA in real time, our work provides clear evidence that ORC preferentially binds nucleosomes—independent of a nearby ACS motif—over non-nucleosomal DNA and, importantly, that this interaction is functionally relevant to MCM DH loading at nucleosomal sites. These findings support the hypothesis illustrated in Fig. 1 that nucleosomes are the dominant directive of origin function in all cells. This demonstration of ARS-sequence independent, but nucleosome-directed origin licensing, is applicable to all eukaryotes including those where a consensus origin sequence has not been found. In this hypothesis, a nucleosome and an adjacent NFR sufficiently wide to accommodate ORC and an MCM DH are the two main prerequisites for pre-RC formation.

The influence of nucleosomes on origin function has been previously investigated[50], but mainly reported as a secondary mechanism to reinforce the DNA sequence-encoded origin specificity by targeting ORC to ARS instead of other DNA sequences occupied by nucleosomes[30,31]. Here we show that nucleosomes—in and of themselves—represent a primary instructive code for replication origins. Given the highly conserved nature of replication initiation factors, we presume that this principle will generalize to higher eukaryotes, just as the many earlier findings in the S.c. system.

A couple of recent studies hinted at the ORC-nucleosome interaction, even though it was not the focus of those studies. A cryo-EM study on the mechanism of MCM DH formation utilized nucleosomes as blocks to MCM DH sliding[12]. While the cryo-EM grid conditions may alter ORC-nucleosome connections, some of the 2D averages did show their proximity indicative of direct interaction. Another single-molecule study showed that, interestingly, RNA polymerase can push ORC and MCM DH to other locations on DNA, even along with a nucleosome, possibly setting the stage for origins to be placed at new positions[51].

Investigating the nature of ORC-nucleosome interaction is a high priority for future studies. The bromo-adjacent homology (BAH) domain of Orc1 is known to bind core histones[32]. BAH deletion alters origin profiles in yeast even though replication still proceeds and cells are viable[52,53]. However, these studies did not demonstrate that the Orc1 BAH domain is the only contact point of ORC to nucleosomes. Indeed, it was shown that BAH-deleted ORC still avidly binds nucleosomes[33], suggesting that there are more unidentified ORC-nucleosome contacts. While it is appealing to study ORC mutants in which most if not all ORC-nucleosome contacts are disrupted, this information is not yet available, and we plan to explore them using our single-molecule assay in the future. For example, mutations of the

Orc4 IH will be interesting to test given that they alter origin patterns on yeast chromatin[22–24].

### NFR adjacent to a nucleosome enables pre-RC assembly independent of the ARS sequence

The second prerequisite for pre-RC formation in our model is the presence of a sufficiently wide NFR that can accommodate the pre-RC complex. It is known that an NFR flanked by regularly positioned nucleosomes is a pronounced feature of S.c. ARS origin sites[15,16]. Our single-molecule data suggest that an MCM DH can be loaded onto DNA lacking an ARS sequence as long as it is adjacent to a nucleosome. ARS may be involved in creating some of these NFRs[46], but is not a necessity. Indeed, our genomic analyses show that most S.c. ORC binding within an NFR occurs over sequences lacking a strong ACS element. There exist numerous non-ARS NFRs in S.c. chromosomes located at 5' and 3' ends of genes. These NFRs are 80–300 bp long and flanked by well-positioned nucleosomes[54–56]. Importantly, recent studies have shown that mutations in the Orc4 IH that abrogate the ability of S.c. ORC to bind canonical ACS alter the genome-wide origin firing pattern such that initiation occurs at open chromatin with wide NFRs (such as promotors) or at novel sequences that may be recognized by the mutated ORC[23,24], supporting the notion that ACS-free NFRs that border a nucleosome can in principle be utilized as an origin site.

On the other hand, the requirement for an NFR permissive to helicase loading excludes many genomic regions from becoming origin sites, such as tightly packaged heterochromatin prevalent in eukaryotic chromosomes. This view explains why replication initiation cannot occur at just any nucleosome-NFR boundary. In addition, other DNA-binding factors may occupy the NFR near the flanking nucleosomes, preventing ORC binding.

### Why have some yeast species evolved DNA sequence-specific origins?

Our model (Fig. 1) infers that there is no fundamental difference in the origin selection mechanism between yeast and higher eukaryotes, but simply that some yeast species (such as S.c.) have evolved a dependence on ARS sequences, perhaps to increase fitness. Yet S.c. cells contain some origins that simply lack a recognizable ACS for ORC[15]. This evolutionary pressure may be related to the high gene density in organisms such as *S. cerevisiae* with a small genome size. The slight advantage that the ACS element confers on yeast ORC binding to non-nucleosomal DNA, along with the strong binding of ORC to a nucleosome adjacent to an ARS sequence, may serve to place replication origins within intergenic regions to help avoid replication-transcription conflicts and genome instability[24]. As such, ARSs add an additional layer of "security" to prevent spurious origin firing. Nevertheless, our results here show that the S.c. system still retains the nucleosome-directed origin licensing capability that is likely also used by higher eukaryotes. Metazoans, which have much larger genomes, may have evolved other mechanisms to circumvent or tolerate transcriptional interference, and thus need not have evolved sequence-specific origins[57].

### A potential explanation for the MCM paradox

Our proposed model that any NFR might enable MCM DH formation mediated by ORC-nucleosome interaction in G1 phase suggests a rather "sloppy" process of origin licensing. It follows that many more sites may be licensed than can be used in S phase for origin firing. While we do not expect replication initiation to take place in every NFR in each cell cycle—because of additional unknown requirements (e.g., histone modifications, MCM requirements to mature to CMG helicases)—it is still possible that many NFRs in the genome allow MCM DHs to be assembled but not normally used. Thus, our findings may offer a mechanistic basis for the "MCM paradox" that refers to the excess MCMs distributed in the genome compared to actively used origins[14]. One may question whether the extra MCMs can be those binding nonspecifically to chromatin or are actually in the form of pre-RC MCM DHs. The first cryo-EM study of the MCM DH provides evidence for the latter scenario[58]. In that study, sufficient MCM DHs were obtained by DNase treatment of yeast chromosomal DNA without protein overexpression for high-resolution structural analysis of the MCM DH. Hence, one may infer a natural abundance of MCM DH on normal yeast chromosomes. These unused "dormant" origins may become activated in the event of DNA damage or other cellular stress that limits the ability of replication forks to progress[59].

There are further steps in S phase that prompt the maturation of MCM DH to dual CMG replicative helicases[30], some of which may involve elements of ARS sequences that can recruit nucleosome remodeling enzymes. Moreover, epigenetic features of the chromatin, such as histone variants and posttranslational modifications—which were missing in our current study and thus not required for functional binding of nucleosomes to ORC to facilitate pre-RC formation—may nonetheless contribute to fine-tuning the "nucleosome origin code" by making particular nucleosomes better or worse binding partners for ORC[60,61].

In conclusion, our work provides a model for how ORC specifies eukaryotic replication origins in general and an explanation for how an excess of MCM DHs can be loaded onto DNA in G1 phase. Future studies, including those using various ORC mutants, and ATPγS, are needed to delineate the detailed mechanism (such as 1-ORC vs. 2-ORC and the OCCM intermediate) of MCM DH loading at nucleosomes. In addition, these single-molecule experiments should also be conducted in diverse eukaryotic systems, including human, to fully elucidate the regulatory mechanisms for nucleosome-ORC-MCM interaction and function.

## Methods

### Protein expression and purification

**S6-ORC.** A recombinant strain of *S. cerevisiae* co-expressing the six subunits of ORC, having a 12-aa S6 tag for fluorescent labeling at the N terminus of Orc1, was constructed as follows. We used either an Orc1 gene with an N-terminal 3× Flag (wt ORC) or further modified the Orc1 subunit gene by insertion of DNA encoding the "S6" peptide (GDSLSWLLRLLN) at the N terminus (S6-ORC)[62]. The six subunits of ORC complex were cloned into integration vectors having the galactose inducible Gal1/10 bidirectional promotor for induction by galactose and were cloned and integrated into the genome of strain OY001 (*ade2-1 ura31 his311,15 trp11 leu23,112 can1100 bar1Δ, MAT*a *pep4øKANMX6*), a strain constructed from W303[63]. The order of integration was: (1) genes encoding 3×Flag-Orc1 or S6-3×Flag-Orc1, and Orc3 (both cloned into pRS404/GAL); (2) genes encoding Orc4 and Orc5 (both cloned into pRS405/GAL); (3) the gene encoding Orc2 (cloned into pRS403/GAL); and (4) the gene encoding Orc6 (cloned into pRS402/GAL). The wt and S6-ORC overexpression strains were constructed by integrating the expression plasmids described above in the yeast genome. Both wt ORC and S6-ORC were purified by the same procedure below.

One liter of S6-ORC cells were grown under selection at 30 °C in SC glucose, then split into 12 2-l fluted flasks, each containing 1 l of YP-glycerol media and grown to an optical density at 600 nm ($OD_{600}$) of 0.4 at 30 °C, arrested with α-factor (50 μg/l) for 2 h, and then induced for 3 h upon addition of 20 g/l of galactose. Cells were harvested by centrifugation, resuspended in a minimal volume of 20 mM HEPES pH 7.6, 1.2% polyvinyl pyrrolidone, and protease inhibitors (Sigma-Aldrich #5056489001) and 0.5 mM PMSF, then snap frozen by dripping into liquid nitrogen. Purification of the S6-ORC complex was performed by lysis of 12 l of frozen cells using two SPEX cryogenic grinding mills (6970 EFM). Ground cells were thawed and debris removed by centrifugation (43,146 × *g* in an SS34 rotor for 2 h at 4 °C); then the supernatant was applied to a 1-ml anti-Flag M2 affinity column (Sigma)

equilibrated in buffer H (50 mM HEPES pH 7.5, 250 mM potassium glutamate, 1 mM EDTA, 10% glycerol). Elution was in buffer H containing 0.15 mg/ml 3× Flag peptide (EZBiolab). Peak fractions were diluted with 2 volumes of buffer H and loaded onto a 1-ml SP HP column (GE Healthcare) and washed with buffer C (50 mM HEPES pH 7.5, 100 mM KOAc). Elution was with an 8-ml linear gradient of 100–600 mM KOAc in 50 mM HEPES pH 7.5. The S6-ORC complex eluted at ~400 mM KOAc. Protein concentration of column fractions was determined by Bradford reagent (Bio-Rad), and stored at −80 °C.

**S6-MCM.** Recombinant S.c. strains that co-expressed the six subunits of Mcm2-7 complex having an N-terminal S6-3×Flag tag for fluorescent labeling (S6-MCM), or only a 3×Flag tag on Mcm3 (wt MCM) were constructed as follows. The six subunits of Mcm2-7 were cloned into integration vectors, each having the galactose inducible Gal1/10 bidirectional promotor for induction by galactose. The vectors were then integrated into the genome of strain OY001. Genes encoding Mcm4 and 3×Flag-S6-Mcm3 (or 3×Flag-Mcm3) were cloned into pRS404/GAL (Trp); genes encoding Mcm6 and Mcm7 were cloned into pRS405/GAL (Leu); the gene encoding Mcm2 was cloned into pRS403/GAL (His); and the Mcm5 gene was cloned into pRS406/GAL (Ura). The vectors were integrated in OY001 in the order described above to yield S6-MCM (or wt MCM). S6-MCM cells were grown and induced as described for S6-ORC. Purification of S6-MCM complex was performed by lysis of 12 l of frozen cells with a SPEX cryogenic grinding mill (6970 EFM). Ground cells were thawed and debris removed by centrifugation ($23,719 \times g$ in a SLC-1500 rotor at 4 °C). The clarified extract was applied to a 3-ml anti-Flag M2 affinity column (Sigma) equilibrated in buffer M [50 mM HEPES pH 7.5, 100 mM potassium glutamate, 2 mM DTT, 1 mM EDTA, 10 mM Mg(OAc)₂, 0.5 mM ATP, 10% glycerol]. The column was washed with 25 ml buffer M, then eluted with buffer M containing 0.15 mg/ml 3× Flag peptide (EZBiolab). Protein concentration of column fractions was determined by Bradford reagent (Bio-Rad), then aliquoted, snap frozen in liquid N₂ and stored at −80 °C.

**Cdc6.** An *E. coli* optimized gene sequence for expression of S.c. Cdc6 in *E. coli* (a generous gift of Dr. Megan Davey, Western University, Canada) was cloned into pET11 (Novagen). The Cdc6 gene was then subcloned into pGEX-6P-1 (GeneScript) to provide an N-terminal GST tag with a PreScission protease site which we refer to as pGST-Cdc6 PST/P. *E. coli* BL21(DE3) cells were transformed with the pGST-Cdc6 PST/P plasmid and cells were grown in LB + 100 μg/ml ampicillin at 37 °C with shaking until reaching an OD₆₀₀ of 0.49, at which time the culture temperature was quickly reduced to 14 °C by shaking in an ice bath. Then IPTG (1 mM) and 0.2% ethanol were added to induce GST-Cdc6 expression for 24 h at 15 °C. Cells were harvested by low speed centrifugation at 4 °C, and the cell pellet was resuspended in buffer G (50 mM Tris-HCl pH 7.5, 1 mM EDTA, 10% glycerol) plus 30 μM spermidine and 500 mM NaCl. Cells were lysed by French Press and the lysate was clarified by centrifugation at $20,000 \times g$ for 1 h at 4 °C. The supernatant was loaded onto a 5-ml glutathione column (GE Healthcare), followed by washing with 20 column volumes of buffer G plus 300 mM NaCl. Elution was performed using 25 ml of 20 mM Tris-HCl pH 8.0, 1 mM EDTA, 10% glycerol, 47 mM glutathione and 300 mM NaCl. Fractions of 1 ml were collected and analyzed by SDS PAGE. Fractions containing GST-Cdc6 were treated with 200 U PreScission protease (Thermo Fisher) for 2 h on ice, then were diluted with buffer A (20 mM HEPES pH 7.5, 1 mM EDTA, 10% glycerol) to a conductivity of 85 uS/CM and loaded onto an 8-ml SP-Sepharose column (Sigma). Cdc6 was eluted in 20-ml steps of buffer A containing either 0.2 M, 0.3 M, 0.4 M, 0.5 M NaCl. Fractions were analyzed for Cdc6 by SDS PAGE, and fractions containing Cdc6 were pooled and passed through a 2-ml GST column to remove remaining GST contaminants. Protein concentration was determined by Bradford reagent (Bio-Rad), then aliquoted, snap frozen in liquid N₂ and stored at −80 °C.

**Cdt1.** Cdt1 was cloned into a pET16b vector followed by replacement of the NcoI/NdeI region with insertion of a DNA segment encoding a 3× Flag tag. *E. coli* was transformed with the pFlag-Cdt1-pET plasmid and cells were grown in LB + 100 μg/ml ampicillin at 37 °C with shaking until reaching an OD₆₀₀ of 0.6, at which time the culture temperature was quickly reduced to 15 °C by shaking in an ice bath. Then IPTG (1 mM) was added to induce Cdt1 expression for 10 h at 15 °C. Cells were then harvested by low speed centrifugation at 4 °C, and the cell pellet was resuspended in 50 ml of buffer B (50 mM HEPES pH 7.5, 1 mM EDTA, 2 mM DTT, 2 mM MgCl₂, 10% glycerol) plus 800 mM NaCl, and lysed by French Press. Cell lysate was clarified by centrifugation at $20,000 \times g$ for 1 h at 4 °C. The supernatant was treated with 1.5 ml Flag beads (Sigma) for 1 h with gentle agitation, then packed into a 5-ml C column (Cytiva) equilibrated in buffer B at 4 °C. Cdt1 was eluted with buffer B containing 175 mM NaCl and 0.2 mg/ml 3× Flag peptide. The preparation was then diluted two-fold with buffer B to reduce conductivity, applied to a 1-ml Heparin agarose column (Sigma), and eluted with a 10-ml linear gradient of 100 mM NaCl to 500 mM NaCl in buffer B. Protein concentration of column fractions was determined by Bradford reagent (Bio-Rad). Peak fractions containing Cdt1 were analyzed in an 8% SDS PAGE, pooled, then aliquoted, snap frozen in liquid N₂ and stored at −80 °C.

**Nap1.** S.c. Nap1 was expressed in *E. coli* from the gene inserted into pGEX-6P-1, a kind gift of Dr. Aaron Johnson (University of Colorado, Denver). The GST-Nap1 was purified essentially as described[64,65], except for passage of the final prep through a GST column (Thermo Fisher). Briefly, the pGEX-GST-Nap1 expression plasmid was transformed into *E. coli* BL21 (DE3) cells, and 6 l were grown in LB plus 100 μg/ml ampicillin to an OD₆₀₀ of 0.5 at which time the culture temperature was quickly reduced to 15 °C by shaking in an ice bath. Then IPTG (1 mM) was added to induce Nap1 expression for 10 h at 15 °C. Cells were harvested by low speed centrifugation at 4 °C, and the cell pellet was resuspended in 100 ml of PBS (Thermo Fisher) plus 500 mM NaCl, 1.5 mM DTT, 1 mM EDTA, 30 mM spermidine and 0.5% Triton X-100. Cells were lysed by French Press, and the cell lysate was clarified by centrifugation. The supernatant was loaded onto a 4-ml GST column (GE Healthcare) equilibrated in PBS containing 1 mM EDTA, 0.1 mM DTT, 0.1 mM PMSF, 0.5% Triton X-100, 10% glycerol, and 350 mM NaCl. Elution was with 50 mM Tris-HCl pH 8.0, 1 mM EDTA, 5 mM DTT, 300 mM NaCl, 10% glycerol and 47 mM glutathione. Fractions of 2 ml were collected and analyzed by SDS PAGE for presence of Nap1. Fractions containing Nap1 were pooled and 100 U of PreScission protease (Thermo Fisher) was added prior to dialysis overnight against buffer C (20 mM Tris pH 7.5, 1 mM EDTA, 1 mM DTT, 150 mM NaCl, 10% glycerol). Twenty eight ml of the dialysate was then loaded onto a 1-ml MonoQ column and eluted with a 20-ml gradient from 150 mM to 1 M NaCl in buffer C. Peak fractions containing Nap1 were pooled and then passed over a GST column to remove any contaminating GST tag and GST-PreScission protease. Protein was analyzed by SDS PAGE and the concentration of column fractions was determined by Bradford reagent (Bio-Rad). Peak fractions of Nap1 were pooled, then aliquoted, snap frozen in liquid N₂ and stored at −80 °C.

**Histone octamer.** Recombinant *Xenopus laevis* and S.c. histones and their mutants were purified as previously described[66]. Briefly, histones were expressed in BL21 (DE3) cells. Histones were extracted from inclusion bodies under denaturing conditions and purified through Q FF and SP FF columns (GE Healthcare). Histone octamers were then refolded by dialysis and purified by gel filtration using a Superdex 200 10/300 GL column.

### Fluorescent labeling of proteins
To label S6-ORC and S6-MCM, the protein, Sfp synthase and dye-CoA (dye = Cy3 or LD650) were incubated at a 1:2:5 molar ratio for 1 h at

room temperature in the presence of 10 mM $Mg^{2+}$. After labeling for 1 h, a ten-fold volume of labeling buffer (for S6-ORC: 50 mM HEPES pH 7.5, 100 mM KOAc, 250 mM KGlu, 1 mM EDTA, 10% glycerol, and 10 mM $MgSO_4$; for S6-MCM: 50 mM HEPES KOH pH 7.5, 100 mM KOAc, 2 mM DTT, 10 mM Mg(OAc)₂, 0.5 mM ATP, and 10% glycerol) was added to dilute the sample. Then the sample was pooled into a prewet 100-kDa Amicon centrifugal filter unit and exchanged with labeling buffer at least five times to remove Sfp and free dye. The final products were aliquoted, flash frozen, and stored at −80 °C.

To label histones, the single-cysteine construct H2A$^{K120C}$ was generated by site-directed mutagenesis. All histones were purified and labeled as previously described[66]. Briefly, they were incubated with Cy3 maleimide (GE Healthcare), or A488 $C_5$ maleimide (Thermo Fisher) at 1:5 molar ratio in a labeling buffer (20 mM Tris-HCl pH 7.0, 7 M Guanidine-HCl, 5 mM EDTA, and 1.25 mM TCEP) for 4 h at room temperature. Labeling reactions were quenched with 80 mM β-mercaptoethanol. Excess dyes were removed by dialysis in a buffer containing 20 mM Tris-HCl pH 7.0, 7 M Guanidine-HCl, and 1 mM DTT. The labeling efficiency varies among batches, ranging from ~50% to >90%.

### Preparation of DNA template for single-molecule experiments

To generate λ$_{ARS1}$ DNA, a 501-bp DNA fragment containing the 185-bp Stillman minimum ARS1 sequence[67] (Supplementary Table 1) was amplified from the yeast chromosome (S288C_ChrIV BK0069382: 462,279–462,787) and inserted into λ DNA (Roche, Cat# 11558706910) with XhoI and NheI restriction enzymes (New England BioLabs). The product was then packaged into phage particles using phage extract (MaxPlax, Epicentre). Plaques were generated on LE392 *E. coli* bacterial lawns (Epicentre) and screened for the ARS1 insert. A screened plaque was used as a phage source to purify λ$_{ARS1}$ DNA by lytic growth[68]. The final λ$_{ARS1}$ DNA is 47,822 bp in length and the ARS1 site is located 33,499–33,999 bp from the left end of the the phage genome. To create a terminally biotinylated λ$_{ARS1}$ DNA, the 12-base 5′ overhang on each end was filled in with a mixture of unmodified and biotinylated nucleotides by the exonuclease-deficient DNA polymerase I Klenow fragment (New England BioLabs). The reaction was conducted by incubating 10 nM λ$_{ARS1}$ DNA, 33 μM each of dGTP/dATP/biotin-11-dUTP/biotin-14-dCTP (Thermo Fisher), and 5 U Klenow in 1× NEB2 buffer at 37 °C for 45 min, followed by heat inactivation at 75 °C for 20 min. The DNA was then ethanol precipitated overnight at −20 °C in 2.5× volume cold ethanol and 300 mM NaOAc pH 5.2. Precipitated DNA was recovered by centrifugation at 20,000 × g at 4 °C for 15 min. After removing the supernatant, the pellet was air-dried, resuspended in TE buffer (10 mM Tris-HCl pH 8.0, 1 mM EDTA) and stored at 4 °C.

### Single-molecule experiments

**Data acquisition.** Single-molecule experiments were performed at room temperature on a LUMICKS C-Trap instrument as previously described[36]. Laminar-flow-separated channels 1–3 were used to form DNA tethers between two 4.35-μm streptavidin-coated polystyrene beads (Spherotech). Channels 4 and 5 served as protein loading and imaging channels. A488, Cy3, and Cy5/LD650 fluorophores were excited by 488, 532 and 638 nm laser lines, respectively. Kymographs were generated via confocal line scanning through the center of the two beads. The tethers were held at a force below 2.5 pN for all experiments, and there is no difference in observable results at low forces of 1–2.5 pN.

**Nucleosome assembly in situ.** Optical traps tethering a single DNA were moved to a channel containing 1 nM of fluorescently labeled S.c. or X.l. histone octamers and 2 nM Nap1 in HR buffer [30 mM Tris-OAc pH 7.5, 20 mM Mg(OAc)₂, 50 mM KCl, 1 mM DTT, 40 μg/ml BSA], and incubated under a tension below 1 pN until a few fluorescent spots were seen. The tether was then moved to Channel 3 containing 0.5 mg/

ml salmon sperm DNA (Thermo Fisher) in HR buffer except for MCM/ORC salt stability assays, in which Channel 3 contained a high-salt buffer [50 mM HEPES pH 7.5, 40 μg/ml BSA, 2 mM DTT, 10 mM Mg(OAc)₂, 500 mM NaCl, 2.5 mM ATP]. A microfluidic flow was turned on for 1 min to gently remove free histones and free Nap1.

**Visualization of ORC.** Optical traps tethering a single bare DNA or nucleosomal DNA were moved to a separate channel containing 2 nM Cy3- or LD650-ORC and 5 nM Cdc6 in an imaging buffer containing 25 mM Tris-OAc pH 7.5, 5% glycerol, 40 μg/ml BSA, 3 mM DTT, 2 mM TCEP, 0.1 mM EDTA, 10 mM Mg(OAc)₂, 50 mM potassium glutamate, and 2.5 mM ATP. The imaging buffer was supplemented with an ATP-regeneration system [60 mg/ml creatine phosphokinase (Sigma) and 20 mM phosphocreatine (Sigma)], a triplet-state quenching cocktail [1 mM cyclooctatetraene (Sigma), 1 mM 4-nitrobenzyl alcohol (Sigma) and 1 mM Trolox (Sigma)], as well as an oxygen scavenging system [10 nM protocatechuate-3,4-dioxygenase (Sigma) and 2.5 mM protocatechuic acid (Sigma)]. Kymographs were typically recorded for 4–10 min.

**Visualization of MCM.** For two-color MCM/nucleosome experiments, a tethered nucleosomal DNA loaded with Cy3 or A488-labeled octamers was moved to a channel containing 10 nM LD650-MCM and 14 nM Cdt1, with or without 2 nM ORC and 5 nM Cdc6 in the imaging buffer described above in the presence of 5 mM ATP. For MCM DH loading experiments, a tethered nucleosomal DNA was moved to a channel containing 10 nM Cy3-MCM, 10 nM LD650-MCM, 4 nM unlabeled ORC, 10 nM Cdc6, 28 nM Cdt1, and 5 mM ATP. For MCM/ORC salt stability experiments, the tether was moved to a channel containing the high-salt buffer described above. The oxygen scavenging system was omitted from the imaging buffer for the MCM photobleaching experiments.

**Data analysis.** Single-molecule force and fluorescence data from the .h5 files generated from C-Trap experiments were analyzed using tools in the lumicks.pylake Python library supplemented with other Python modules in a custom GUI Python script titled "C-Trap.h5 File Visualization GUI", which was written to extract confocal images and traces.

### Genomic analysis

We analyzed the following published datasets: SRR034475 and SRR034476 (Orc1 ChIP-seq)[15]; SRR1261333 (Mcm2-7 ChIP-seq)[40]; GSE147927 (ChIP-exo data for Orc1, Abf1, and Reb1)[43]; GSM2589911 (MNase data)[44]. The ChIP-seq data were aligned to yeast genome (version Scer3) with bowtie2, and peaks were called using MACS2 (threshold: effective *p* value 0.01). Orc1 and Mcm2-7 ChIP peaks tend to be broader than typical transcription factors, probably due to the large size of the complex. As a result, multiple peaks tend to appear as clusters within a few hundred bps. We wrote a MATLAB algorithm to select a single peak with the largest area within each cluster. We found 295 Orc1 peaks in total (sorted based on the total intensity of the Orc1 ChIP-seq signal in the ±1 kb region in Fig. 7a). ARS annotation was downloaded from the Saccharomyces Genome Database. All Orc1 peak locations (peak center ±100 bp) and the corresponding ARSs are shown in Supplementary Data 1.

To get the Orc1 motif information, we extracted the genomic sequences within peak center ±500 bp. Using the PWM in ref. 42, we used a MATLAB program to scan through the sequence and calculated the PWM score for each 17-bp window on both strands. At each base, PWM score calculated the $\log_2$ of probability of the appearance of a base divided by the probability for that base to appear in the genome. For example, if for Position X, "A" appears 90% of time, while in the genome it is 25%, then the score of Position X is $\log_2(0.9/0.25)$. The final score is the sum of scores for all positions in the element. Abf1 and Reb1 motifs were analyzed the same way, except that we scanned

through shorter sequences (peak center ±150 bp) because the ChIP peaks of these factors are narrower.

## Statistics and reproducibility

Errors reported in this study represent the standard deviation. $p$ values were determined from unpaired two-tailed $t$-tests using GraphPad Prism 9 (ns, not significant; $*p < 0.05$; $**p < 0.01$; $***p < 0.001$; $****p < 0.0001$). All experiments were independently repeated at least three times with similar results. Representative results are shown in figures.

## Reporting summary

Further information on research design is available in the Nature Research Reporting Summary linked to this article.

## Data availability

The data that support this study are available from the corresponding authors upon reasonable request. Publicly available sequencing datasets used in this study can be found in: SRR034475 and SRR034476, SRR1261333, GSE147927, GSM2589911. Source data are provided with this paper.

## Code availability

The home-written script "C-Trap.h5 File Visualization GUI" is available on the LUMICKS Harbor platform (https://harbor.lumicks.com/single-script/c5b103a4-0804-4b06-95d3-20a08d65768f). All custom codes can be made available upon reasonable request.

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

## Acknowledgements
We thank X. Zhao (Sloan Kettering Institute) for critical reading of the manuscript, B. Stillman (Cold Spring Harbor Laboratory) for feedback on a preprint version of this manuscript, L. Langston, R. Mayle, G. Schauer, N. Yao, and D. Zhang in the O'Donnell Laboratory for some of the reagents, J. Watters in the Liu Laboratory for data analysis codes, and other members of the Liu and O'Donnell groups for discussions. L.B. is supported by National Institutes of Health (R01GM118682 and R35GM139654). M.E.O. is supported by NIH (R01GM115809) and the Howard Hughes Medical Institute. S.Liu is supported by the Robertson Foundation, the Alfred P. Sloan Foundation, the Pershing Square Sohn Cancer Research Alliance, and an NIH Director's New Innovator Award (DP2HG010510).

## Author contributions
S.Liu and M.E.O. oversaw the project. M.R.W. prepared the DNA templates. O.Y. prepared the yeast expression constructs for protein production, and purified the factors. M.R.W., O.Y. and S.Li prepared and labeled the replication proteins. S.Li prepared the nucleosome samples and performed the biochemical assays. S.Li and M.R.W. performed the single-molecule experiments. L.B. performed the genomic data analysis. S.Liu, M.E.O., S.Li and L.B. wrote the manuscript.

## Competing interests
The authors declare no competing interests.
