## [Peer Review File · Nature Communications]

REVIEWER COMMENTS

Reviewer #1 (Remarks to the Author):

In this manuscript submitted by Li et al., the authors used an in vitro real time visualization on tethered ARS1-containing DNA template to monitor the dynamics of labeled MCM-DH and histones at different conditions. The authors demonstrated that recruitment of budding yeast ORC to chromatin DNA for MCM-DH, despite the well-documented ACS-B1 specificity, is also highly influenced by the presence of reconstituted nucleosomes. They showed that ScORC can recruit MCM-DH to NFRs without an ARS. They extrapolate their observations to higher eukaryotes where no clear ACS-like motif in replication origins has been identified for those ORCs. Hence, nucleosome-dependent recruitment across genome might be a universal mechanism of all eukaryotic ORCs. They speculated that budding yeast ORC has acquired an ACS-B1 specificity to avoid transcription-replication conflicts given the small genome size. In general, their data and observations are sound and self-consistent. This work shows convincingly that ORC has an affinity for nucleosomes on DNA that is relevant for ORC recruitment and MCM loading - a notion harbored by many but only demonstrated explicitly in this study. Some of their observations support previous reports on the mobilities of the licensing complex and may serve as additional support to those observations.

Comments:

1. Are the DH loaded by ARS1 and nucleosomes at non-ARS regions functionally the same, despite their difference in high-salt resistance (Fig.3I)? The authors might want to expand on the explanation of their differences. Is it because of difference in loading process of DH?
2. It is noted that most of the DH assembly experiments in this report were done with 50mM salt throughout, instead of the 100mM or 150mM salt, which are the usual conditions in previous reports (e.g. Yeeles et al 2015). The authors should exclude the possibility that their observations are due to the buffer condition used in this study. This may help explain the state of the salt sensitive ORC-MCM DH described here, which were not observed in prior studies.
3. It would be most interesting to include Δ BAH and Δ IH mutant of ORC as a control to show direct ORC-nucleosome interactions account for the observation of ARS-independent recruitment.

Minor issues:

1. Line 181-184, why does ScORC have stronger affinity to XI nucleosomes than Sc nucleosomes if they share high sequence homology?

2. Line 282, reference 20 does not support "...“humanizing mutations” of S.c. ORC that abrogate its ability to bind ACS change the genome-wide origin pattern such that initiation occurs at NFRs lacking ACS.....” because these mutants may have acquired a new ACS.

Bik Tye

Reviewer #2 (Remarks to the Author):

This manuscript describes novel findings concerning the mechanism of eukaryotic replication origin licensing. *S. cerevisiae* origins generally contain consensus ARS for ORC binding. How ORC binding is specified in most other eukaryotes, which do not have consensus sequences, is not well understood. The authors reconstitute origin licensing with fluorescent *S. c* ORC and MCM on single DNAs in an optical trap. On naked DNA, Cy3-ORC binds stably only at ARS. LD650-MCM is loaded in an ORC-dependent fashion, with more stable MCM DH formation at ARS. The authors reconstitute *S.c* and *X. l* nucleosomes on trapped DNAs and show ORC and MCM frequently bind nucleosomes from both species, independently of ARS. The single-molecule data is complemented with an analysis of existing *S. c* genomic data showing frequent ORC binding to non-consensus ARS. Together, this data is used to propose a model of origin specifying where ORC binds nucleosomes adjacent to a nucleosome free region. In yeast, consensus ARS might help create the nucleosome free region but this consensus ARS is not essential as nucleosomes can also bind ORC. Higher eukaryotes lack consensus ARS so might use nucleosome-ORC interactions to specify origins. This is a thought provoking finding that warrants future work to investigate (1) the nature of nucleosome-ORC interactions and (2) whether nucleosomes specify human origins in a similar manner.

General comments:

1. The work presented here warrants future investigation of ORC-nucleosome interaction. There should be further discussion of the potential nature of the ORC-nucleosome interaction – for instance by expanding on the findings by De Ionnas et al. (2019) and Müller et al. (2010). The Orc1 BAH – H4 N-terminus interaction is structurally characterised in De Ionnas et al. (2019). To support their conclusion that ORC is recruited to nucleosomes, the authors could show ORC-histone interaction can be mutationally disrupted.

2. The authors use two-step photobleaching and high salt wash to show MCM DH formation. Scherr et al. (2022) have shown high MCM mobility after licensing in high salt. Sanchez et al. (2021), however, show slow MCM diffusion after high salt wash. As an alternative, as shown in both these previous papers, the authors should show licensing in ATPγS results in formation of OCCM rather than MCM DHs.

3. The authors provide strong evidence of ORC-nucleosome interaction, but this should be contrasted with other findings. Miller et al. (2019) found a yeast nucleosome proximal to the ACS does not enhance licensing in reconstituted reactions. In addition, Scherr et al. (2022) used reconstituted single-molecule licensing reactions with yeast proteins and present no evidence of nucleosome-ORC colocalization. In this paper, nucleosomes were reconstituted by salt dialysis rather than using Nap1. The authors should show nucleosome-ORC colocalization is not changed if nucleosomes are assembled by salt dialysis.

4. The key finding of the structural study Miller et al. (2019) was to reconcile licensing mechanisms involving a single ORC (Ticau et al. (2015)) and multiple ORCs (Coster and Diffley (2017)). This structural work showing recruitment of a 2nd MCM involves a different ORC-MCM interaction to the first ORC has been further supported by single-molecule evidence from Gupta et al. (2021). This recent work should be introduced in a paper describing reconstituted origin licensing. The authors should make clear a caveat of Figure 1 is not incorporating an 'ORC flip' or two ORC model of DH recruitment.

5. Fig 7 – in general for the explanation of the re-analysed genomic data presented here, the authors need to be more explicit how their findings provide new insight. They show many ARS are non-consensus but was this not already known? They show ORCs binding a non-consensus ARS “seem to locate near a nucleosome-NFR junction” but this is not quantified. There is also no discussion whether NFR with low ORC/MCM occupancy exist and why this might be.

Specific comments:

1. To help future studies, it would be useful to see fluorescence scanned SDS-PAGE gels for the proteins used in this study. On line 485 more details of the buffer exchange method used should be included.

2. Line 100: although the results are largely consistent with previous results, Sanchez et al. (2021) contrastingly show Cdc6 is not required for ARS specificity.

3. In many of the figures, green/red colour use is not accessible, unless a merged image is shown alongside separate channel images.

4. Line 119: When referring to Figure 3G/H/I in text, it is not clear two-colour MCM DHs are being used here.

5. Line 140: the ORC-nucleosome colocalization in Figure 4B is clear and there is more ORC binding than on naked DNA in Figure 2C. Is it possible to directly compare the efficiency of ORC binding to DNA in these experiments?
6. Fig 5D – Only ~25% of MCMs bind in proximity to nucleosomes. In contrast >80% of ORC is bound to nucleosomes (Fig 4C). This should be explained further in the text.
7. Fig 6C – ~30% of MCMs are described as stably bound after HSW. There are no clear examples of stably bound MCMs after HSW in Fig 6B or Supp. Fig 8.
8. Sup. Fig. 8B – the sliding of dual-colour labelled MCM is not clear – if one of the fluorophores has bleached this should be indicated.
9. Fig 7B – either title two columns of origin examples with low and high PWN scores or show the individual PWN scores for these origins in the figure.
10. Line 243: “binds to nucleosomes near an NFR” is an overstatement from the genomic data presented here – there is no direct evidence of nucleosome binding from this part of the manuscript.
11. Line 290 – why would later origin firing be explained by slower origin licensing?
12. Could the authors comment on the forces used (in methods section) – is DNA held below 1 pN for all experiments?

Reviewer #3 (Remarks to the Author):

Review of Li et al “Nucleosome-directed replication origin licensing independent of a consensus DNA sequence” for Nature Communications. In this manuscript, the authors have characterized the loading of the MCM double hexamer by ORC at nucleosomes with adjacent free DNA. The main advance in understanding is that nucleosomes localize ORC binding at adjacent free DNA regions to efficiently load MCM. They tested this model in *S.c.* where there are well described origin regions and conclude that ACS elements are not strictly required for preRC formation and instead create a NFR region, where ORC can bind. The authors conclude that ORC binding to free nucleosomes to load MCM double hexamers at

NFRs is a more generally conserved licensing paradigm in all eukaryotes. The manuscript is very well written and described, however I have a few comments that may need to be addressed.

1) Does the nucleosome have a directionality of ORC interaction and therefore MCM loading? For example, can MCM sliding give insight into the asymmetrical interaction of ORC with a nucleosome?

2) As 30% of MCMs are mobile, 30% are stable at ORC (i.e. strong engagement), and I presume the other 40% dissociate. Can you say anything about which of these cases include MCM double hexamers? For example, are double hexamers more prone to sliding and single hexamers have more strong engagement with ORC or vice versa. Or maybe single hexamers are not fully loaded and dissociate?

3) Is it possible to disrupt interaction of ORC with nucleosomes and test whether ORC binds preferentially at ACS sequences?

4) To show conservation which is a central conclusion of this study, CHIP-seq should be reanalyzed in other organisms. Maybe even Pombe where ACS sequences are less likely?

5) Does the ARS sequence itself preclude nucleosome assembly at that site in favor of others? Or is there any indication that the sequence adjacent to the ARS site is a stronger nucleosome assembly/positioning sequence?

RESPONSE TO REVIEWER COMMENTS

Reviewer #1 (Remarks to the Author):

In this manuscript submitted by Li et al., the authors used an in vitro real time visualization on tethered ARS1-containing DNA template to monitor the dynamics of labeled MCM-DH and histones at different conditions. The authors demonstrated that recruitment of budding yeast ORC to chromatin DNA for MCM-DH, despite the well-documented ACS-B1 specificity, is also highly influenced by the presence of reconstituted nucleosomes. They showed that ScORC can recruit MCM-DH to NFRs without an ARS. They extrapolate their observations to higher eukaryotes where no clear ACS-like motif in replication origins has been identified for those ORCs. Hence, nucleosome-dependent recruitment across genome might be a universal mechanism of all eukaryotic ORCs. They speculated that budding yeast ORC has acquired an ACS-B1 specificity to avoid transcription-replication conflicts given the small genome size. In general, their data and observations are sound and self-consistent. This work shows convincingly that ORC has an affinity for nucleosomes on DNA that is relevant for ORC recruitment and MCM loading - a notion harbored by many but only demonstrated explicitly in this study. Some of their observations support previous reports on the mobilities of the licensing complex and may serve as additional support to those observations.

Response: We appreciate the reviewer for her concise and accurate summary of the scientific advance that is made by this report.

Comments:

1. Are the DH loaded by ARS1 and nucleosomes at non-ARS regions functionally the same, despite their difference in high-salt resistance (Fig.3I)? The authors might want to expand on the explanation of their differences. Is it because of difference in loading process of DH?

Response: We apologize for causing this confusion. In Fig. 3I we compare MCM behavior at ARS1 vs non-ARS bare DNA sites (no nucleosome). The MCM behavior at nucleosomes is reported in Fig. 6C, which actually shows a very similar distribution to the one for ARS1 DNA (Fig. 3I). While a rigorous proof needs further investigation, this similarity in high-salt resistance indicates that MCM DHs at ARS1 DNA and at nucleosomes are loaded through the same process and are functionally equivalent. We have added a sentence on this point to the revised manuscript (Lines 198-201).

2. It is noted that most of the DH assembly experiments in this report were done with 50mM salt throughout, instead of the 100mM or 150mM salt, which are the usual conditions in previous reports (e.g. Yeeles et al 2015). The authors should exclude the possibility that their observations are due to the buffer condition used in this study. This may help explain the state of the salt sensitive ORC-MCM DH described here, which were not observed in prior studies.

Response: We appreciate this comment. Most, or all, studies of MCMs show a significant number that form on DNA in 100 mM glutamate fall off DNA at 500 mM NaCl. Essentially, glutamate is not a strong dissociative salt like NaCl. Therefore, we agree that we might observe more non-specific binding at 50 mM glutamate compared to use of 100 mM glutamate, but that the important thing is that we employed the stringent use of 0.5 M NaCl to identify truly loaded MCMs from non-specific binding. In other words, the significant fraction of MCMs that did not dissociate at 0.5 M NaCl in our experiments suggests DHs were successfully assembled in the glutamate buffer that we used.

3. It would be most interesting to include Δ BAH and Δ IH mutant of ORC as a control to show direct ORC-nucleosome interactions account for the observation of ARS-independent recruitment.

Response: We appreciate this comment. The genetics literature shows that removal of BAH from Orc1 still supports apparently normal cell viability¹, so one should expect this mutant to support MCM DH assembly. Indeed, Hizume et al. (2013) showed that ORC ^{Δ BAH} still binds nucleosomes in vitro². The deletion of Orc4 IH will be extremely interesting to test given that it alters origin patterns on yeast chromatin³⁻⁵. These are important experiments for future studies. We have added this to the revised Discussion (Lines 298-306).

At present we are in the process to make protein preps of the ORC mutants along the lines the reviewer indicates. However, the fact that the two leading authors have left the lab, and that it took them 3 years to develop sufficient expertise to conduct the single-molecule experiments described in this paper, new experiments with mutants will take a significant amount of time and develop into a separate paper.

Minor issues:

1. Line 181-184, why does ScORC have stronger affinity to XI nucleosomes than Sc nucleosomes if they share high sequence homology?

Response: We made this speculation based on the fraction of ORC dissociation from S.c. vs X.I. nucleosomes at high salt observed in our single-molecule experiments. In reality, the difference could be confounded by other factors such as histone labeling efficiency and nucleosome loading density. Without a detailed biochemical characterization of the binding affinities (not the focus of this study), we have toned down this conclusion in the revised manuscript as follows (Lines 185-187):

“While it appears that X.I. nucleosomes preserve S.c. ORC better than S.c. nucleosomes, this could be due to subtle differences in the experimental conditions rather than a meaningful difference in their binding affinity.”

2. Line 282, reference 20 does not support “...“humanizing mutations” of S.c. ORC that abrogate its ability to bind ACS change the genome-wide origin pattern such that initiation occurs at NFRs lacking ACS.....” because these mutants may have acquired a new ACS.

Response: We thank the reviewer for pointing this out. We have modified the text as follows (Lines 316-321):

“Importantly, recent studies have shown that mutations in the Orc4 IH that abrogate the ability of S.c. ORC to bind canonical ACS alter the genome-wide origin firing pattern such that initiation occurs at open chromatin with wide NFRs (such as promoters) or at novel sequences that may be recognized by the mutated ORC^{3,4}, supporting the notion that ACS-free NFRs that border a nucleosome can in principle be utilized as an origin site.”

Bik Tye

Reviewer #2 (Remarks to the Author):

This manuscript describes novel findings concerning the mechanism of eukaryotic replication origin licensing. *S. cerevisiae* origins generally contain consensus ARS for ORC binding. How ORC binding is specified in most other eukaryotes, which do not have consensus sequences, is not well understood. The authors reconstitute origin licensing with fluorescent *S. c* ORC and MCM on single DNAs in an optical trap. On naked DNA, Cy3-ORC binds stably only at ARS. LD650-MCM is loaded in an ORC-dependent fashion, with more stable MCM DH formation at ARS. The authors reconstitute *S.c* and *X. I* nucleosomes on trapped DNAs and show ORC and MCM frequently bind nucleosomes from both species, independently of ARS. The single-molecule data is complemented with an analysis of existing *S. c* genomic data showing frequent ORC binding to non-consensus ARS. Together, this data is used to propose a model of origin specifying where ORC binds nucleosomes adjacent to a nucleosome free region. In yeast, consensus ARS might help create the nucleosome free region but this consensus ARS is not essential as nucleosomes can also bind ORC. Higher eukaryotes lack consensus ARS so might use nucleosome-ORC interactions to specify origins. This is a thought provoking finding that warrants future work to investigate (1) the nature of nucleosome-ORC interactions and (2) whether nucleosomes specify human origins in a similar manner.

Response: We thank the reviewer for their positive evaluation on our work and fully agree on the directions of future work in this area.

General comments:

1. The work presented here warrants future investigation of ORC-nucleosome interaction. There should be further discussion of the potential nature of the ORC-nucleosome interaction – for instance by expanding on the findings by De Ionnas et al. (2019) and Müller et al. (2010). The Orc1 BAH – H4 N-terminus interaction is structurally characterised in De Ionnas et al. (2019). To support their conclusion that ORC is recruited to nucleosomes, the authors could show ORC-histone interaction can be mutationally disrupted.

Response: We fully agree that investigating the nature of ORC-nucleosome interaction is a high priority for future studies. As the reviewer noted, the BAH domain of Orc1 is well known to bind core histones⁶ and BAH deletion alters origin profiles in yeast even though replication still proceeds and cells are viable^{1,7}. However, neither study demonstrated that BAH is the only contact point of ORC to nucleosomes. Indeed, Hizume et al. showed that BAH-deleted ORC still avidly binds nucleosomes², suggesting that there are more unidentified ORC-nucleosome contacts. While we would very much like to disrupt most if not all ORC-nucleosome contacts and test the constructs in our single-molecule assays, this information is not yet available, and we plan to explore it in future work. We now include a discussion on this point in the revised manuscript (Lines 298-306).

2. The authors use two-step photobleaching and high salt wash to show MCM DH formation. Scherr et al. (2022) have shown high MCM mobility after licensing in high salt. Sanchez et al. (2021), however, show slow MCM diffusion after high salt wash. As an alternative, as shown in both these previous papers, the authors should show licensing in ATPγS results in formation of OCCM rather than MCM DHs.

Response: We appreciate these suggestions. Indeed, ATPγS was strategically used to demonstrate the existence of the OCCM intermediate towards the formation of MCM DH. In our study, we focused our experiments (which are time consuming and technically challenging) on the locations of origin licensing rather than the detailed pathway of MCM DH formation. Therefore, we felt that the photobleaching analysis and high-salt wash experiments are sufficient to support our conclusion that MCM DHs are formed at ARS1, non-ARS1, and nucleosomes bound to non-ARS DNA. We have added a note at the end of the revised manuscript that further experiments, including those using ATPγS, are needed to delineate the detailed mechanism (such as 1-ORC vs 2-ORC, the OCCM intermediate, etc.) of MCM DH loading at nucleosomes (Lines 368-370).

Regarding the different MCM DH mobility reported by previous papers, we are not sure about its exact source, although we suspect that it is caused by slightly different buffer conditions and sample preps.

3. The authors provide strong evidence of ORC-nucleosome interaction, but this should be contrasted with other findings. Miller et al. (2019) found a yeast nucleosome proximal to the ACS does not enhance licensing in reconstituted reactions. In addition, Scherr et al. (2022) used reconstituted single-molecule licensing reactions with yeast proteins and present no evidence of nucleosome-ORC colocalization. In this paper, nucleosomes were reconstituted by salt dialysis rather than using Nap1. The authors should show nucleosome-ORC colocalization is not changed if nucleosomes are assembled by salt dialysis.

Response: We appreciate this comment. As the reviewer is aware, direct ORC-nucleosome interaction is a widely reported phenomenon by in vitro studies^{2,6,8}. Regarding the two papers mentioned by the reviewer, the Miller et al. paper is a structural study that identified the MO complex as an intermediate to PreRC. This report used the nucleosome as a block of MCM DH sliding and did not focus on nucleosome-ORC interaction. The Scherr et al. paper cites our bioRxiv preprint related to this manuscript and does not mention any inconsistency with our work. We likewise see no inconsistencies (the Scherr et al report has one figure that involves nucleosomes (Fig. 4)). Interestingly, even under their conditions of low protein population on DNA, their data show an instance of ORC-nucleosome co-localization. They may have other examples, but this was not the main point of their paper. Regardless, it is important to note that we do not claim that every ORC targets a nucleosome, but that a measurable percentage do. We now discuss our work in the context of previous literature (Lines 290-297) and also cite the Scherr et al. paper in the revised Discussion for the fact that RNAP can push MCM DH to new locations which may redefine origins in a cell.

For nucleosome formation we used Nap1 because it makes our experiments more streamlined and less time-consuming (they remain technically challenging). We used the salt dialysis method to form nucleosomes

in our initial experiments and obtained indistinguishable results (see an example kymograph below showing ORC-nucleosome colocalization).

4. The key finding of the structural study Miller et al. (2019) was to reconcile licensing mechanisms involving a single ORC (Ticau et al. (2015)) and multiple ORCs (Coster and Diffley (2017)). This structural work showing recruitment of a 2nd MCM involves a different ORC-MCM interaction to the first ORC has been further supported by single-molecule evidence from Gupta et al. (2021). This recent work should be introduced in a paper describing reconstituted origin licensing. The authors should make clear a caveat of Figure 1 is not incorporating an ‘ORC flip’ or two ORC model of DH recruitment.

Response: We agree with the reviewer. While our report does not touch upon the argument of whether one or two ORCs form a MCM DH, we have added a sentence in the Introduction (Lines 35-37) and a note in the caption of Figure 1 to emphasize that recent reports demonstrate that MCM DH is loaded via an ‘ORC-flip’ or ‘two-ORC’ mechanism. These additions to the revised manuscript cite all the references mentioned above.

5. Fig 7 – in general for the explanation of the re-analysed genomic data presented here, the authors need to be more explicit how their findings provide new insight. They show many ARS are non-consensus but was this not already known? They show ORCs binding a non-consensus ARS “seem to locate near a nucleosome-NFR junction” but this is not quantified. There is also no discussion whether NFR with low ORC/MCM occupancy exist and why this might be.

Response: The traditional view of ARS in budding yeast is that they contain a 17-bp motif TTT(T/A)TTTAT(A/G)TTT(T/A)G(G/T)T. Although it is known that this motif is degenerate and most origins contain mismatches to the consensus, it is still thought that a partial match to this consensus motif is essential for origin function⁹. Here, by analyzing the ORC binding vs. ACS PWM score (with additional analysis of ACS motifs in genome-wide NFRs), we provided a quantitative assessment of that statement. We show that 1) only 20% of the ARS contain a passable ACS consensus (score > 11.9), and this percentage is significantly lower than some other yeast transcription factors with their own specific binding motifs (such as Abf1 and Reb1 as mentioned in the text). 2) The ACS motif plays a strong role in directing ORC binding ONLY when it is a near-perfect match to the consensus. For example, seven out of eight ACS with a score > 14 are occupied by Orc1. For strong motifs, Orc1 binding correlates with the motif score (Pearson correlation coefficient $R = 0.51$). 3) Most Orc1 binding occurs over sequences with a PWM score between 2 and 11.9. In these cases, the ACS motif only plays a minor role in ORC binding ($R = 0.11$).

We agree with the reviewer that non-consensus ARS “seem to locate near a nucleosome-NFR junction” is an overstatement. Due to the limited resolution of ChIP-seq and quality of the ChIP-exo data, it is impossible to calculate the distance between Orc1 binding site to the nearest nucleosome. We therefore changed the text as follows (Lines 249-252):

“Orc1 ChIP peaks over sequences without a consensus ACS motif (PWM score < 9) also contain NFRs (Figure 7B). In these cases, ORC potentially bind to nucleosome-NFR junctions, although higher resolution data are needed to resolve the exact ORC binding location relative to nucleosomes.”

There are four to five thousand NFRs in the yeast genome¹⁰ but less than 300 Orc1 binding sites, which means that most NFRs do not serve as origins. The reviewer raised an important question as to why that is the case. Here we offer some speculations. First, there are only ~1,500 copies of Orc1 protein per yeast cell (SGD). Therefore, ORC will bind selectively to the NFRs that offer high affinities. ACS motifs provide some specificity when they closely match the consensus as shown above. However, a high affinity may also arise from certain DNA shape as indicated by previous modeling work¹¹. Depending on the footprint of ORC, some short NFRs may not provide enough space for ORC to bind. In addition, other DNA-binding factors may occupy the NFR near the flanking nucleosomes, preventing ORC from binding. Finally, histones with certain modifications may also modulate ORC binding affinity. These points are now made more explicitly in the revised manuscript (Lines 244-258, 316-326) and the plot above is now included as Supplementary Fig. 10.

Specific comments:

1. To help future studies, it would be useful to see fluorescence scanned SDS-PAGE gels for the proteins used in this study. On line 485 more details of the buffer exchange method used should be included.

Response: We have added the buffer exchange details to the Methods section (Lines 521-526) and the fluorescence scan of SDS-PAGE gels to Supplementary Fig. 2.

2. Line 100: although the results are largely consistent with previous results, Sanchez et al. (2021) contrastingly show Cdc6 is not required for ARS specificity.

Response: We agree with the reviewer that it is puzzling as to why Sanchez et al (2021) found that Cdc6 is not required for ARS specificity, as it is contrary to the many reports (including ours) that Cdc6 is indeed required. We do not know how to explain their unusual result and expect that it will be explained in their future publications. We have removed this reference from this sentence (but still cited in other contexts).

3. In many of the figures, green/red colour use is not accessible, unless a merged image is shown alongside separate channel images.

Response: To better illustrate colocalization of different colors, we have shown separate channel images or windows with individual lasers off in the kymographs.

4. Line 119: When referring to Figure 3G/H/I in text, it is not clear two-colour MCM DHs are being used here.

Response: We now clarify the text as follows (Lines 119-122):

“To obtain further evidence for MCM DH formation, we used a mixture of Cy3-labeled and LD650-labeled MCMs. After ORC-mediated MCM binding, we moved the DNA tether into a separate channel containing a

high-salt buffer (0.5 M NaCl), upon which we observed high mobility of dual-colored MCM (red and green) (Figure 3G & 3H).”

5. Line 140: the ORC-nucleosome colocalization in Figure 4B is clear and there is more ORC binding than on naked DNA in Figure 2C. Is it possible to directly compare the efficiency of ORC binding to DNA in these experiments?

Response: We appreciate the reviewer’s comment. However, it is not straightforward to directly compare the efficiency of ORC binding in these two experiments, one on bare DNA (Fig. 2) while the other containing nucleosomes (Fig. 4). The experiments used different ORC samples (Cy3-labeled vs. LD650-labeled) and different incubation times. Therefore, even though we agree that there is more ORC binding on nucleosomal DNA than on bare DNA (as expected given ORC’s affinity to nucleosomes), we refrained from drawing a definitive conclusion.

6. Fig 5D – Only ~25% of MCMs bind in proximity to nucleosomes. In contrast >80% of ORC is bound to nucleosomes (Fig 4C). This should be explained further in the text.

Response: We appreciate this comment. The two figure panels measure different things but are not contradictory. In Fig. 4C, we measure ORC on nucleosomes at the ARS1 site vs. at other regions of DNA and find that ARS does not make a significant difference. The plot reports the fraction of stably bound ORC under one condition (i.e., at an ARS or non-ARS nucleosome), but not its distribution between nucleosomes and non-nucleosomal DNA. In Fig. 5D, we measure the fraction of MCM bound at nucleosomes vs. non-nucleosomal DNA. These are now clearly stated in the figure captions.

7. Fig 6C – ~30% of MCMs are described as stably bound after HSW. There are no clear examples of stably bound MCMs after HSW in Fig 6B or Supp. Fig 8.

Response: We now provide an example of stably bound (non-sliding) MCM after high-salt wash in Supplementary Fig. 8D (also shown below). We thank the reviewer for this suggestion.

8. Sup. Fig. 8B – the sliding of dual-colour labelled MCM is not clear – if one of the fluorophores has bleached this should be indicated.

Response: We thank the reviewer for bringing this up. Indeed, the red fluorophore seemed to have bleached before high-salt wash, as shown by the second “green laser off” window. This is now indicated in the caption of Supplementary Fig. 8. We have separated individual color channels and added a zoomed-in view of the high-salt section of the kymograph to more clearly show MCM sliding.

9. Fig 7B – either title two columns of origin examples with low and high PWN scores or show the individual PWN scores for these origins in the figure.

Response: We added “high score ACS motif” and “low score ACS motif” to the two columns.

10. Line 243: “binds to nucleosomes near an NFR” is an overstatement from the genomic data presented here – there is no direct evidence of nucleosome binding from this part of the manuscript.

Response: Please see our reply to General comment #5 above.

11. Line 290 – why would later origin firing be explained by slower origin licensing?

Response: We thank the reviewer and agree that later origin firing in S phase is not explained by slower licensing in G1 phase. We have thus removed this postulation.

12. Could the authors comment on the forces used (in methods section) – is DNA held below 1 pN for all experiments?

Response: We conducted all experiments at a force at or below 2.5 pN. We have compared results obtained at 2.5 pN with those at 1 pN and did not see a difference. We now mention this in the revised Methods (Lines 570-572).

Reviewer #3 (Remarks to the Author):

Review of Li et al “Nucleosome-directed replication origin licensing independent of a consensus DNA sequence” for Nature Communications. In this manuscript, the authors have characterized the loading of the MCM double hexamer by ORC at nucleosomes with adjacent free DNA. The main advance in understanding is that nucleosomes localize ORC binding at adjacent free DNA regions to efficiently load MCM. They tested this model in S.c. where there are well described origin regions and conclude that ACS elements are not strictly required for preRC formation and instead create a NFR region, where ORC can bind. The authors conclude that ORC binding to free nucleosomes to load MCM double hexamers at NFRs is a more generally conserved licensing paradigm in all eukaryotes. The manuscript is very well written and described, however I have a few comments that may need to be addressed.

Response: We thank the reviewer for their positive summary and evaluation on our work, and we appreciate their comments to improve this manuscript.

1) Does the nucleosome have a directionality of ORC interaction and therefore MCM loading? For example, can MCM sliding give insight into the asymmetrical interaction of ORC with a nucleosome?

Response: This is a very interesting question. We have done some analysis on the directionality of ORC-nucleosome interaction and MCM sliding as shown below. Due to ambiguity of the tether direction, we can only examine tethers having ARS1-bound ORC/MCM. For 10 examples at ARS1, we did not see any obvious direction bias, suggesting that ORC interaction and MCM loading can occur from either direction. However, considering the small sample size, we do not feel that the data is sufficient to present in a publication.

2) As 30% of MCMs are mobile, 30% are stable at ORC (i.e. strong engagement), and I presume the other 40% dissociate. Can you say anything about which of these cases include MCM double hexamers? For example, are double hexamers more prone to sliding and single hexamers have more strong engagement with ORC or vice versa. Or maybe single hexamers are not fully loaded and dissociate?

Response: The presumption from past literature is that MCMs that are not stable on DNA are OCCM-like, or maybe single hexamers, and that MCM DHs are stable and can slide on DNA. Thus, we agree with the reviewer that the 40% that dissociate were single hexamers or OCCM-type recruitment that did not fully proceed to MCM DH. Our photobleaching analysis and two-color MCM experiments support this notion, with the caveat of imperfect labeling efficiency. Many studies have in fact observed similar findings—that many MCMs on DNA dissociate in high salt and only the remaining population are double hexamers. Examples include Figs 2 and 3 in Evrin et al (2009)¹², Fig. 1A and Fig. 3 A,C,D in Remus et al (2009)¹³.

3) Is it possible to disrupt interaction of ORC with nucleosomes and test whether ORC binds preferentially at ACS sequences?

Response: We would be very keen on doing this experiment. Unfortunately, we are not aware of a way to disrupt the ORC-nucleosome interaction. The Orc1 BAH domain is the only known connection to the nucleosome, but BAH deletion does not disrupt nucleosome-ORC binding². Thus, other unidentified connections of ORC to nucleosomes must exist. We do not yet know how many connections there are and what mutations are needed to fully disrupt ORC-nucleosome interaction, which we feel is beyond the scope of this study. We have added sentences about this point to the revised Discussion (Lines 298-306) and at the final paragraph as a future direction (Lines 368-370).

4) To show conservation which is a central conclusion of this study, CHIP-seq should be reanalyzed in other organisms. Maybe even Pombe where ACS sequences are less likely?

Response: We appreciate the reviewer's comment. However, there are no other model eukaryotic organisms that utilize a defined origin sequence, or ACS consensus. The unique aspect of ARS/ACS in budding yeast enabled the ChIP-seq analysis, but we are unable to perform a parallel analysis in another eukaryotic organism.

5) Does the ARS sequence itself preclude nucleosome assembly at that site in favor of others? Or is there any indication that the sequence adjacent to the ARS site is a stronger nucleosome assembly/positioning sequence?

Response: We appreciate this insightful comment. S.c. ARS/ACS contains polyA/T tracks that are known to have lower intrinsic affinity towards histones¹⁴. We have shown that ORC binding can further lower local nucleosome occupancy¹⁵. Our computational study of nucleosome positioning disfavors strong nucleosome positioning sequences in the native yeast genome¹⁰. Instead, strong nucleosome positioning is generated by pioneer factors and remodeling complexes.

References cited in the response

1. Muller, P. et al. The conserved bromo-adjacent homology domain of yeast Orc1 functions in the selection of DNA replication origins within chromatin. *Genes Dev* **24**, 1418-33 (2010).
2. Hizume, K., Yagura, M. & Araki, H. Concerted interaction between origin recognition complex (ORC), nucleosomes and replication origin DNA ensures stable ORC-origin binding. *Genes Cells* **18**, 764-79 (2013).
3. Hu, Y. et al. Evolution of DNA replication origin specification and gene silencing mechanisms. *Nat Commun* **11**, 5175 (2020).
4. Lee, C.S.K. et al. Humanizing the yeast origin recognition complex. *Nat Commun* **12**, 33 (2021).
5. Li, N. et al. Structure of the origin recognition complex bound to DNA replication origin. *Nature* **559**, 217-222 (2018).

6. De Ioannes, P. et al. Structure and function of the Orc1 BAH-nucleosome complex. *Nat Commun* **10**, 2894 (2019).
7. Bell, S.P., Mitchell, J., Leber, J., Kobayashi, R. & Stillman, B. The multidomain structure of Orc1p reveals similarity to regulators of DNA replication and transcriptional silencing. *Cell* **83**, 563-8 (1995).
8. Kuo, A.J. et al. The BAH domain of ORC1 links H4K20me2 to DNA replication licensing and Meier-Gorlin syndrome. *Nature* **484**, 115-9 (2012).
9. Nieduszynski, C.A., Knox, Y. & Donaldson, A.D. Genome-wide identification of replication origins in yeast by comparative genomics. *Genes Dev* **20**, 1874-9 (2006).
10. Kharerin, H. & Bai, L. Thermodynamic modeling of genome-wide nucleosome depleted regions in yeast. *PLoS Comput Biol* **17**, e1008560 (2021).
11. Li, W.-C., Deng, E.-Z., Ding, H., Chen, W. & Lin, H. iORI-PseKNC: A predictor for identifying origin of replication with pseudo k-tuple nucleotide composition. *Chemometrics and Intelligent Laboratory Systems* **141**, 100-106 (2015).
12. Evrin, C. et al. A double-hexameric MCM2-7 complex is loaded onto origin DNA during licensing of eukaryotic DNA replication. *Proc Natl Acad Sci U S A* **106**, 20240-5 (2009).
13. Remus, D. et al. Concerted loading of Mcm2-7 double hexamers around DNA during DNA replication origin licensing. *Cell* **139**, 719-30 (2009).
14. Struhl, K. & Segal, E. Determinants of nucleosome positioning. *Nat Struct Mol Biol* **20**, 267-73 (2013).
15. Yan, C., Chen, H. & Bai, L. Systematic Study of Nucleosome-Displacing Factors in Budding Yeast. *Mol Cell* **71**, 294-305 e4 (2018).

REVIEWERS' COMMENTS

Reviewer #1 (Remarks to the Author):

I am satisfied with the revised manuscript. The authors have addressed all of my concerns.

Reviewer #2 (Remarks to the Author):

Please see attached

Reviewer #3 (Remarks to the Author):

The authors have done a reasonable job addressing both my concerns as well as those from the other reviewers. Although there are still some questions on the interactions of ORC with the adjacent nucleosome for positioning purposes, this work will set the foundation to determine that specificity.